# Overcoming attenuation bias in regressions using polygenic indices

Hans van Kippersluis [1,2] ✉, Pietro Biroli [3], Rita Dias Pereira[1,2],
Titus J. Galama [1,2,4,5], Stephanie von Hinke [1,2,6], S. Fleur W. Meddens[1,7],
Dilnoza Muslimova [1,2], Eric A. W. Slob [1,8,9], Ronald de Vlaming [2,4] &
Cornelius A. Rietveld [1,2,9]

Measurement error in polygenic indices (PGIs) attenuates the estimation of their effects in regression models. We analyze and compare two approaches addressing this attenuation bias: Obviously Related Instrumental Variables (ORIV) and the PGI Repository Correction (PGI-RC). Through simulations, we show that the PGI-RC performs slightly better than ORIV, unless the prediction sample is very small ($N < 1000$) or when there is considerable assortative mating. Within families, ORIV is the best choice since the PGI-RC correction factor is generally not available. We verify the empirical validity of the simulations by predicting educational attainment and height in a sample of siblings from the UK Biobank. We show that applying ORIV between families increases the standardized effect of the PGI by 12% (height) and by 22% (educational attainment) compared to a meta-analysis-based PGI, yet estimates remain slightly below the PGI-RC estimates. Furthermore, within-family ORIV regression provides the tightest lower bound for the direct genetic effect, increasing the lower bound for the standardized direct genetic effect on educational attainment from 0.14 to 0.18 (+29%), and for height from 0.54 to 0.61 (+13%) compared to a meta-analysis-based PGI.

Genome-wide association studies (GWASs) have firmly established that, with few exceptions, most human (behavioral) traits are highly "polygenic"—that is, influenced by many individual genetic variants, each with a very small effect size[1,2]. A natural consequence of this has been the widespread adoption of so-called polygenic indices (PGIs), weighted sums aggregating the small effects of numerous genetic variants (single-nucleotide polymorphisms (SNPs)), which enable out-of-sample genetic prediction of complex traits[3–5]. It is common practice to meta-analyze GWAS summary statistics from as many samples as possible to foster the identification of genome-wide significant SNPs[1]. Through the law of large numbers, this strategy has also proven

to be very effective in reducing measurement error in the PGI and thus to boost the power and accuracy for analyses involving PGIs[4]. PGIs are now able to explain a non-negligible proportion of the variance in health and behavioral traits[6].

Empirically constructed PGIs are nonetheless a noisy proxy for the true latent PGI (i.e., the "additive SNP factor," defined as the best linear predictor of the phenotype from the measured genetic variants[6]) because, amongst other reasons, the GWAS underlying the construction of the PGI is based on a finite sample[7,8]. The noise in the GWAS coefficients translates into noisy measures of the PGI and leads to what is typically known as "attenuation bias" (i.e., a bias towards zero) in the

[1]Erasmus School of Economics, Erasmus University Rotterdam, Rotterdam, The Netherlands. [2]Tinbergen Institute, Amsterdam, The Netherlands. [3]Department of Economics, University of Bologna, Bologna, Italy. [4]School of Business and Economics, Vrije Universiteit Amsterdam, Amsterdam, The Netherlands. [5]Center for Social and Economic Research, University of Southern California, Los Angeles, CA, USA. [6]School of Economics, University of Bristol, Bristol, UK. [7]Statistics Netherlands, The Hague, The Netherlands. [8]Medical Research Council Biostatistics Unit, Cambridge University, Cambridge, UK. [9]Erasmus University Rotterdam Institute for Behavior and Biology, Rotterdam, The Netherlands. ✉e-mail: hvankippersluis@ese.eur.nl

coefficient of a PGI in a regression. As a result, the predictive power of today's PGIs is still substantially smaller than the SNP-based heritability estimates, which constitute an upper bound for the predictive power of PGIs[9]. For example, the current predictive power, or variance explained in the phenotype ($R^2$), of the educational attainment (EA) PGI is about 12–16%[10], whereas the SNP-based heritability is estimated to be in the range 22–28%[11–13]. Importantly, the returns to increasing GWAS sample size in terms of gaining predictive power of the PGI are rapidly decreasing[14,15]. For example, for a trait with a SNP-heritability of 25%, it takes a sample of ~1 million to construct a PGI explaining 20%, but it would take a 7-fold increase in discovery sample size to achieve an $R^2$ of 24% (see Eq. (1) below with $M \sim 70,000$). Hence, reducing measurement error in the conventional way of meta-analyzing ever-larger discovery samples has rapidly diminishing payoffs.

There is a burgeoning body of papers exploiting PGIs in medicine, biology, and the social sciences. In many applications, the goal is to estimate the regression coefficient of the true latent PGI. Taking the EA PGI as an example, a non-exhaustive list of applications includes (i) studies estimating the pathways through which the EA PGI influences lifetime outcomes including speech and reading skills[16] or wealth accumulation[17]; (ii) studies that estimate direct genetic effects using within-family estimation to understand the mechanisms through which molecular differences translate into education differences[18]; (iii) studies that investigate gene-environment interplay in EA[19,20], (iv) studies that investigate intergenerational persistence of education[21–23]; and (v) studies including a PGI as a control variable to reduce omitted variable bias or improve precision[24]. While there have been important advances in our understanding of how functional priors and optimizing construction methods can enhance the predictive power of a given PGI[25–27], most existing applications still employ a (meta-analysis based) PGI without applying a correction for measurement error.

Recent studies have laid out the advantages of a measurement error correction and either suggested OLS estimates that have undergone some reasonable correction for attenuation[6,28] or IV estimation[29] to deal with measurement error in the PGI. First, DiPrete et al.[29] suggested an instrumental variables (IV) approach to reduce measurement error in the PGI as a by-product of their Genetic Instrumental Variable (GIV) method. The intuition of the IV approach is simple: when we split the GWAS discovery sample into two, we can obtain two PGIs that both proxy the same underlying "true" PGI. For example, when splitting the UK Biobank (UKB) at random into two discovery samples, both resulting PGIs approximate the same true latent PGI. Hence, theoretically, their correlation should be 1. However, in practice, their correlation will be smaller than 1 since the GWAS sample sizes used to construct these PGIs are finite and therefore each PGI will be subject to measurement error. In case (i) of polygenicity, (ii) the sources of measurement error are independent, and (iii) the relative variance of measurement error in the PGI is the same across the two discovery samples, then the correlation between the two PGIs reveals the degree of measurement error. The IV approach in turn uses this information to correct (or "scale") the observed association between the PGI and the outcome. Second, Becker et al.[6] developed an approach to disattenuate estimated effects of the PGI on a trait, on the basis of external information of the trait's SNP-based heritability (see also ref. [28]). Intuitively, in this approach the SNP-based heritability is estimated in a first step, after which the coefficient of the PGI is re-scaled to match this SNP-based heritability in a second step. Since this approach was proposed alongside the introduction of the PGI repository project[6], we will refer to this approach as the PGI repository correction (PGI-RC).

In this study, we use simulations and empirical analyses to compare the IV and PGI-RC approach to reduce attenuation bias in analyses involving PGIs. Our goal is to estimate a coefficient that is free from attenuation bias due to measurement error in a PGI. As such, we are not primarily interested in boosting the out-of-sample predictive power of a given PGI (in terms of e.g., the $R^2$) for which ever-increasing GWAS discovery samples remain important. We are also not primarily interested in estimating the direct (or "causal") effect of a PGI. In between-family designs, PGIs typically capture not only effects of inherited variation (direct effects), but also effects of population stratification, demography, and relatives (so-called indirect genetic effects)[30]. In the simulations, we compare the performance of a meta-analysis based PGI (benchmark) to (1) results obtained using an IV approach[29], which relies on two PGIs constructed from two non-overlapping GWASs, and (2) the PGI repository correction[6]. For the IV approach, we avoid the arbitrary choice of selecting one PGI as the independent variable and the other as IV by using the recently developed Obviously-Related Instrumental Variables (ORIV) technique[31]. In our comparison of the benchmark, ORIV, and PGI-RC we consider various degrees of (i) genetic nurture (i.e., the effect of parental genotype on the child's outcomes), (ii) assortative mating, and (iii) genetic correlation across discovery and prediction samples. In turn, in the empirical application, we compare the benchmark, ORIV, and PGI-RC using data on height and educational attainment from the sibling sample of the UK Biobank[32]. We conclude that ORIV is preferred over the PGI-RC if the prediction sample is very small ($N < 1000$), when there is assortative mating, and in within-family designs. For small discovery samples, or when there is imperfect genetic correlation between the discovery and prediction sample, PGI-RC is the preferred choice.

## Results
### Simulation study
To compare the performance of meta-analysis, ORIV, and the PGI-RC to estimate the standardized effect of the PGI on an outcome variable, we developed a general-purpose Python tool called GNAMES (Genetic-Nurture and Assortative-Mating-Effects Simulator). This tool allows users to efficiently simulate multi-generational genotype and phenotype data under genetic nurture (GN) effects and assortative mating (AM). The GNAMES tool itself, a description of its technical details, and a tutorial, are freely available on the following GitHub repository: https://github.com/devlaming/gnames.

In the simulations, we partition simulated genotype and phenotype data into three sets: two sets with equal sample size to perform non-overlapping GWASs (discovery) and one set to construct PGIs (prediction). For both sets of GWAS results, we construct a separate PGI. In addition, we also construct a PGI based on the meta-analysis of the two GWASs, in line with the common practice of using the largest possible discovery sample to construct a PGI. Since the SNP effect sizes are simulated to be independent, the PGI simply equals the sum of the SNPs, weighted by the respective GWAS coefficients.

In our setup, every family has two children. Importantly, in the GWASs, data for only one sibling per family are used, and so the resulting PGIs are based on between-family GWAS results, as is common in the literature (but see Howe et al.[30] for a recent exception). The outcome of each simulation is a data file with the individual's ID, the father and mother's ID, a simulated outcome, two PGIs constructed from the two non-overlapping GWAS discovery samples, and the meta-analysis PGI. Further details of the simulation design are provided in Supplementary Methods 1.

The simulations are calibrated based on educational attainment (EA), but we show that our conclusions hold for traits with a different level of heritability in Supplementary Results 2. The SNP-based heritability is approximately equal to 25% for EA in most samples[11–13]. Hence, we fix the SNP-based heritability of the outcome at 25%. In turn, we use different settings that vary in terms of:

- The prediction sample—To create realistic variation in prediction sample sizes we vary $N_{\text{prediction}}$ in the range (1000; 2000; 4000; 8000; 16,000). For example, a recent study on EA[33] employs prediction samples of similar sizes (the Dunedin Study ($N = 810$), the Environmental Risk Longitudinal Twin (E-Risk)

Study ($N = 1860$), AddHealth ($N = 5526$), the Wisconsin Longitudinal Study (WLS, $N = 7111$) and the Health and Retirement Study (HRS, $N = 8546$)).

- The size of the GWAS discovery sample—As described in refs. 7,8, the predictive power of a PGI mainly depends on the variance explained by each SNP, and the ratio of the GWAS discovery sample to the number of SNPs. In particular, under the assumption that all SNPs explain an equal proportion of the SNP-based heritability, De Vlaming et al. (Eq. 2)[34] approximate the predicted $R^2$ of a given PGI as

$$R^2(PGI, Y) \approx h_{SNP}^2 \frac{h_{SNP}^2}{\left(h_{SNP}^2 + \left(\frac{M}{N_{GWAS}}\right)\right)}, \quad (1)$$

where $h_{SNP}^2$ is the SNP-based heritability, $M$ is the number of SNPs and $N_{GWAS}$ denotes the GWAS discovery sample size. To maintain a manageable simulation space, we hold the number of SNPs fixed at $M = 5000$, and set $h_{SNP}^2 = 0.25$. In order to simulate realistic levels of predictive power for a PGI, we use $N_{GWAS}$ in the range (2000; 4000; 8000; 16,000; 32,000). These values for the GWAS discovery samples then respectively generate an expected $R^2$ of 2.3% (i.e., close to the PGI performance in the first EA GWAS, EA1,[35]), 4.2% (-EA2,[12]), 7.1%, 11.1% (-EA3,[36]) and 15.4% (-EA4,[10]). To verify that our results are not sensitive to downsizing both $M$ and $N_{GWAS}$ for computational reasons, we also analyze an additional setting with $M = 100,000$ and $N_{GWAS} = 100,000$. The results remain very similar, see Supplementary Results 2.

- The presence of genetic nurture—We hold $h_{SNP}^2$ (i.e., the SNP-heritability) constant at 0.25 in each generation. In the presence of genetic nurture, the estimated SNP-based heritability is however a combination of direct genetic effects and genetic nurture[37]. If the direct genetic effects and genetic nurture components are independent, the heritability is given by: $h_{SNP}^2 = h^2 + 0.5n^2$, where $h^2$ denotes the phenotypic variance accounted for by direct genetic effects, and $n^2$ denotes the phenotypic variance accounted for by genetic nurture. In scenarios without genetic nurture, we fix the SNP-based heritability at $h_{SNP}^2 = h^2 = 0.25$. In scenarios with genetic nurture, we set $h^2 = 0.2$ and $n^2 = 0.1$ such that the genetic nurture is half the size of the direct genetic effect. This parametrization is again loosely following empirical evidence for EA, where the indirect genetic effect represents roughly half of the additive SNP factor[10,18].

- The degree of assortative mating—We vary assortative mating on the outcome variable (i.e., the phenotypic correlation between mates) between 0 and 1 in increments of 0.25. Plausible levels of assortative mating on education vary between 0.1 and 0.6[38–40].

- The genetic correlation—We vary the genetic correlation between the GWAS discovery samples and the prediction sample between 0.25 and 1 in increments of 0.25, where the two GWAS discovery samples have a perfect genetic correlation. An imperfect genetic correlation may for example arise if the discovery GWAS is performed in a UK sample and the prediction is performed in a non-UK sample. Additionally, we vary the genetic correlation between discovery sample 1 and discovery sample 2 from 0.25 to 1 in increments of 0.25, where we maintain a perfect genetic correlation between discovery sample 2 and the prediction sample.

For each scenario, we perform 100 simulation replications ("runs"), and we average the results over these runs. In all scenarios, we estimate (i) a linear regression of the outcome on the meta-analysis PGI; (ii) ORIV, on basis of a two-stage least squares (2SLS) regression

where we simultaneously use both independent PGIs as instrumental variables for each other; and (iii) the PGI-RC where we scale the coefficient obtained under (i) by the estimated SNP-based heritability. The SNP-based heritability is estimated in the prediction sample using MGREML (Multivariate Genome-based Restricted Maximum Likelihood)[41]. In these GREML analyses, individuals with a genetic relatedness larger than 0.05 are excluded.

The default PGI-RC estimator ignores the estimation error in the estimation of the scaling factor (as acknowledged on p.14 of the Supplementary Information of Becker et al.[6]). This may lead to anti-conservative standard errors, and in the univariate case the true standard error is equal to the standard error of the square root of the estimated SNP-heritability. We will therefore present both the default PGI-RC ("PGI-RC (Default)") application, as well as the PGI-RC that incorporates the uncertainty in the scaling factor ("PGI-RC (GREML unc.)"). Details of the estimation procedures can be found in Methods.

Our main evaluation criterion is the resulting point estimate and 95% confidence interval of the estimated coefficient, to compare the bias of a particular method from the known true coefficient. The true coefficient is that of a PGI constructed based on an infinitely large between-family GWAS. However, in order to balance bias as well as precision, in Supplementary Results 1 we additionally present the root mean squared error, which is a function of both the bias as well as the variance.

**Variation in prediction sample size.** Figure 1 shows the coefficient estimates and their 95% confidence intervals for a meta-analysis PGI, ORIV, and the PGI-RC for varying sample sizes of the prediction sample. The estimates derive from our baseline scenario (no genetic nurture, no assortative mating) and where the GWAS sample size is held constant such that the resulting meta-analysis PGI roughly corresponds to EA4 (i.e., $R$-squared ~ 15.4%). Since we simulated a scenario without genetic nurture and assortative mating, the between-family (Fig. 1a) and within-family (Fig. 1b) analyses target the same coefficient (i.e., a SNP-based heritability of 0.25, or a standardized coefficient of 0.5; see "Methods").

Judging from the point estimates, in a between-family setting, ORIV and the PGI-RC clearly outperform a meta-analysis PGI, with limited differences between them in this scenario. An increasing prediction sample size shrinks the confidence intervals but leaves the coefficients largely unaffected. One notable exception is that for a small prediction sample size ($N \leq 1000$), the PGI-RC slightly underestimates the true coefficient. When using a small prediction sample, the uncertainty in the PGI-RC (GREML unc.) is considerably larger than suggested in the default application PGI-RC (default). With a relatively large GWAS sample, the RMSE for ORIV tends to be slightly smaller than for the PGI-RC (see Supplementary Table 2 in Supplementary Results 1).

In a within-family design, the PGI-RC is not available. Confidence intervals are clearly larger within families than between families, since the variation in the PGI is more limited, but again shrink with an increasing prediction sample size. Similar to the between-family analysis, within-family ORIV is consistent and clearly outperforms a meta-analysis based PGI in this scenario.

**Variation in GWAS sample size.** Figure 2 shows the corresponding figure from the same scenario but now holding the prediction sample size constant at $N = 16,000$ and varying the GWAS sample size such that the $R$-squared of the meta-analysis PGI varies from 2.3% (EA1) to 15.4% (EA4). Whereas a meta-analysis PGI clearly benefits from an increased GWAS sample size, for a relatively large prediction sample size of $N = 16,000$ both ORIV and the PGI-RC are approximately unbiased irrespective of the GWAS sample size. The PGI-RC shows a narrower confidence interval compared with ORIV, even when taking into account uncertainty in the GREML estimates, with the difference

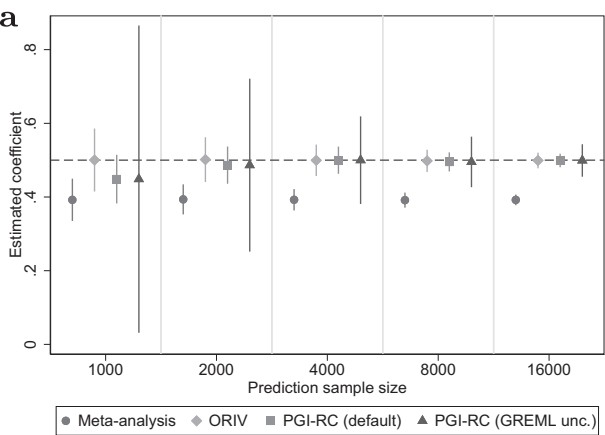 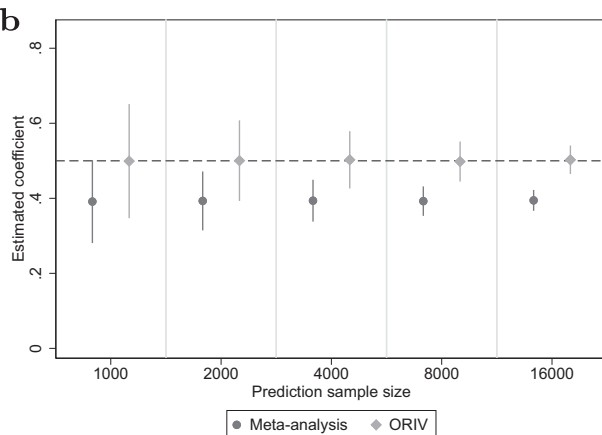

**Fig. 1 | Variation in prediction sample size. a** Between-family analyses. **b** Within-family analyses. Data are presented as the estimated coefficients +/− 1.96 times the standard error (95% confidence interval) for the Polygenic Index (PGI) using meta-analysis (circles), Obviously Related Instrumental Variables (ORIV, rhombuses), the default PGI-RC procedure (squares), and the PGI-RC procedure, taking into account

uncertainty in the GREML estimates (triangles) in the baseline scenario (no genetic nurture, no assortative mating) and holding constant the GWAS discovery sample such that the resulting meta-analysis PGI has an $R^2$ of 15.4%. The horizontal dashed line represents the true coefficient. The simulation results are based on 100 replications.

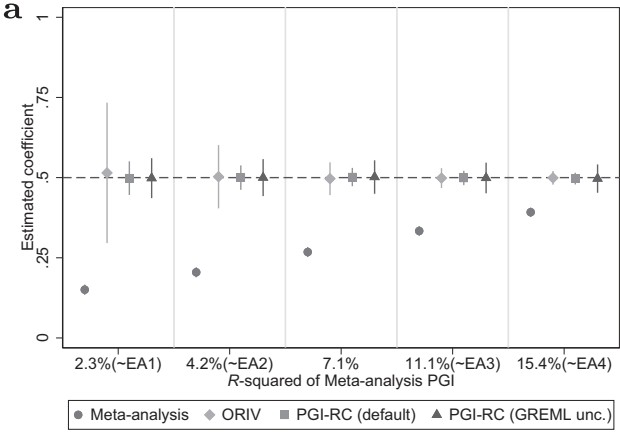 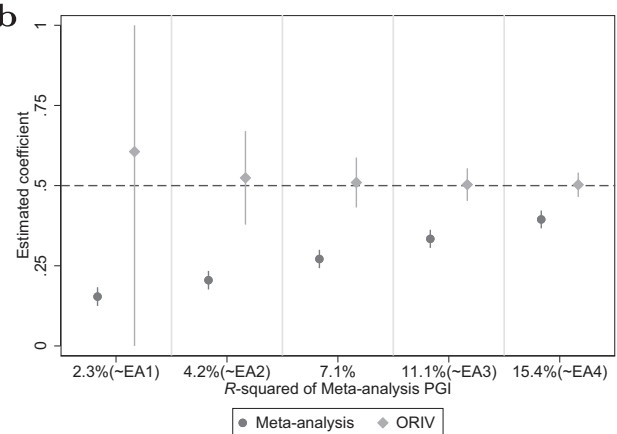

**Fig. 2 | Variation in GWAS sample size. a** Between-family analyses. **b** Within-family analyses. Data are presented as the estimated coefficients +/− 1.96 times the standard error (95% confidence interval) for the Polygenic Index (PGI) using meta-analysis (circles), Obviously Related Instrumental Variables (ORIV, rhombuses), the default PGI-RC procedure (squares), and the PGI-RC procedure, taking into account

uncertainty in the GREML estimates (triangles) in the baseline scenario (no genetic nurture, no assortative mating) and holding constant the prediction sample at $N = 16,000$. The confidence interval for EA1 in the within-family analysis extends beyond the displayed range (−0.08 to −1.29). The dashed line represents the true coefficient. The simulation results are based on 100 replications.

becoming smaller for larger GWAS discovery samples (see also Supplementary Table 3 in Supplementary Results 1).

**Variation in both prediction and GWAS sample size.** In Table 1 we vary both the GWAS as well as the prediction sample size simultaneously. In particular, we compare the performance of a relatively small GWAS sample size, calibrated to be resulting in an $R$-squared of 4.2% (roughly EA2) for different prediction sample sizes, and a relatively large GWAS sample size, calibrated to be resulting in an $R$-squared of 15.4% (roughly EA4), again for different prediction sample sizes. As expected, when the GWAS sample size increases, the coefficient of a meta-analysis PGI comes closer to the true value (0.5), but even for a relatively large GWAS sample size it is still considerably biased. As before, an increasing prediction sample size does not decrease the bias in the coefficient of a meta-analysis PGI, but only shrinks the confidence interval.

ORIV is somewhat biased when both the GWAS sample and the prediction sample are relatively small. This is not surprising, since it is

well known that the bias of IV is inversely proportional to the first-stage $F$-statistic[42,43]. As we derive in the "Methods" section, in this context, the first-stage $F$-statistic is given by (see Eq. (25) for a derivation):

$$F = \frac{corr(PGI_1, PGI_2)^2 (N - 2)}{1 - corr(PGI_1, PGI_2)^2}. \tag{2}$$

Thus, the bias in ORIV is determined by the correlation between the two PGIs, as well as the prediction sample size $N$. The correlation between the two PGIs is determined directly by the $R$-squared of the independent PGIs (see Eqs. (10) and (17) in the "Methods" section). Therefore, with a small GWAS sample size and a small prediction sample size, ORIV is biased. Interestingly, when either the GWAS sample size increases (i.e., moving from EA2 to EA4 while holding constant the prediction sample at $N = 1000$) or the prediction sample size increases (i.e., moving from $N = 1000$ to $N = 4000$ or $N = 16,000$ while holding the GWAS discovery sample constant), ORIV quickly converges to the true point estimate.

**Table 1 | Estimated coefficients for the Polygenic Index (PGI) in the baseline scenario (no genetic nurture, no assortative mating; between-family analyses only)**

| GWAS | Prediction sample | Meta-analysis | ORIV | PGI-RC (Default) | PGI-RC (GREML unc.) |
|------|-------------------|---------------|------|------------------|---------------------|
| ~EA2 | $N = 1000$ | 0.209 | 0.522 | 0.414 | 0.414 |
|      |           | (0.147–0.271) | (0.015–1.029) | (0.287–0.542) | (0.000–0.828) |
| ~EA2 | $N = 4000$ | 0.205 | 0.501 | 0.497 | 0.497 |
|      |           | (0.174–0.237) | (0.294–0.708) | (0.421–0.573) | (0.339–0.655) |
| ~EA2 | $N = 16,000$ | 0.204 | 0.500 | 0.500 | 0.500 |
|      |           | (0.188–0.219) | (0.403–0.598) | (0.462–0.539) | (0.443–0.558) |
| ~EA4 | $N = 1000$ | 0.392 | 0.505 | 0.472 | 0.472 |
|      |           | (0.334–0.450) | (0.417–0.593) | (0.402–0.542) | (0.024–0.920) |
| ~EA4 | $N = 4000$ | 0.386 | 0.497 | 0.500 | 0.500 |
|      |           | (0.357–0.415) | (0.453–0.541) | (0.463–0.538) | (0.381–0.619) |
| ~EA4 | $N = 16,000$ | 0.387 | 0.499 | 0.500 | 0.500 |
|      |           | (0.373–0.402) | (0.477–0.521) | (0.481–0.519) | (0.456–0.544) |

Notes: Data are presented as the estimated coefficients +/− 1.96 times the standard error (95% confidence interval) for the Polygenic Index (PGI) using OLS regression on a meta-analysis based PGI (column 3), a 2SLS regression using ORIV (column 4), the default PGI-RC procedure (column 5), and the PGI-RC procedure, taking into account uncertainty in the GREML estimates (column 6). The GWAS discovery sample is set such that the resulting meta-analysis PGI has an $R^2$ of 4.2% (~EA2) and 15.4% (~EA4). The true coefficient is 0.5. The simulation results are based on 100 replications.

In this baseline scenario, the point estimates of the PGI-RC are largely independent of the GWAS sample size, but are very sensitive to a small prediction sample. The picture that emerges from Table 1 is that the PGI-RC outperforms ORIV when the prediction sample size is relatively large but the GWAS sample size is small; whereas ORIV tends to outperform the PGI-RC when the GWAS sample size is large but the prediction sample is relatively small. In case both the GWAS and prediction samples are small, both methods tend to estimate biased coefficients surrounded by wide confidence intervals, although the RMSE is still comfortably below that of a meta-analysis PGI (see Supplementary Table 4 in Supplementary Results 1).

**Genetic nurture.** When genetic nurture is present, the results (Fig. 3 and Supplementary Table 5 in Supplementary Results 1) very much resemble those obtained in the baseline case (Figs. 1 and 2). The coefficient of a meta-analysis PGI is consistently attenuated; ORIV and the PGI-RC both accurately target the correct point estimate, but confidence intervals are wide for the PGI-RC in small prediction samples, and for ORIV in small discovery samples. Since the results with and without genetic nurture are very similar, we confirm the earlier finding that confounding factors at the GWAS stage do not bias the ORIV and PGI-RC estimates in targeting the additive SNP factor between families[29]. The most important difference is that with genetic nurture, the between- and within-family results are starting to diverge. In within-family designs, ORIV continues to outperform a meta-analysis PGI, yet slightly underestimates the true coefficient because of the noise introduced in the PGI since the GWAS did not control for genetic nurture (see Methods for a more extensive discussion).

**Assortative mating.** Figure 4 shows the results for varying levels of assortative mating. In the top two panels we do not include genetic nurture, while in the bottom two panels we do model genetic nurture in addition to assortative mating. Interestingly, whereas ORIV is consistent in the presence of assortative mating, the PGI-RC estimator is biased and overestimates the true coefficient (see also Supplementary Table 6 in Supplementary Results 1). This is not surprising, as the PGI-RC relies on an estimate of the SNP-based heritability obtained using GREML, which is known to be biased when assortative mating is present[44,45].

The simulations are designed such that the true coefficient does not change with increasing levels of assortative mating (AM). This is achieved by standardizing the true PGI to mean zero and unit variance in each generation of the forward simulation, and then assigning effect

$\sqrt{h^2}$ to this standardized PGI. This approach keeps the "contemporary" heritability fixed and gives a clear target coefficient. Without standardization in each generation, the target coefficient is expected to increase with increasing levels of AM, as the genetic variance and heritability of the trait increase[45]. Similar to our baseline simulation, in Supplementary Results 2 we provide simulation results showing that, also without standardization, ORIV provides consistent estimates, whereas the PGI-RC overestimates the true coefficient.

**Genetic nurture and assortative mating.** In our simulations, we draw independent effect sizes for the direct genetic effects and the genetic nurture components in the founding population, such that the direct genetic and genetic nurture components are independent. However, when modeling genetic nurture and assortative mating simultaneously, over generations, a correlation emerges between the direct genetic effects and the genetic nurture effects (see Supplementary Methods 1 for details). This implies that the SNP heritability and thereby the target coefficient for the PGI are no longer an additive sum of direct genetic effects and genetic nurture effects, but also include a positive covariance among them. This covariance becomes larger when the degree of assortative mating increases, driving up the target coefficient (dashed line in panel c) when assortative mating is stronger. Interestingly, we find that in between-family analyses ORIV continues to target the correct coefficient, yet as before, the PGI-RC systematically overestimates the target coefficient in the presence of assortative mating (see also Supplementary Table 6 in Supplementary Results 1). As expected, performing a within-family analysis (panel d) largely purges the bias induced by genetic nurture and assortative mating, with again ORIV providing a conservative estimate of the direct genetic effect in within-family analyses.

**Imperfect genetic correlation.** Figure 5 shows the sensitivity of the between-family approaches to an imperfect genetic correlation between the GWAS and prediction samples (Fig. 5a) and between two GWAS samples that are meta-analyzed or used for ORIV (Fig. 5b). For a meta-analysis based PGI, these two settings are closely related because an imperfect genetic correlation between two GWAS samples by definition implies an imperfect genetic correlation with the prediction sample. The results show that both a meta-analysis PGI as well as ORIV are very sensitive to an imperfect genetic correlation between the discovery and prediction sample (see also Supplementary Table 7 in Supplementary Results 1 for the corresponding RMSE values). In

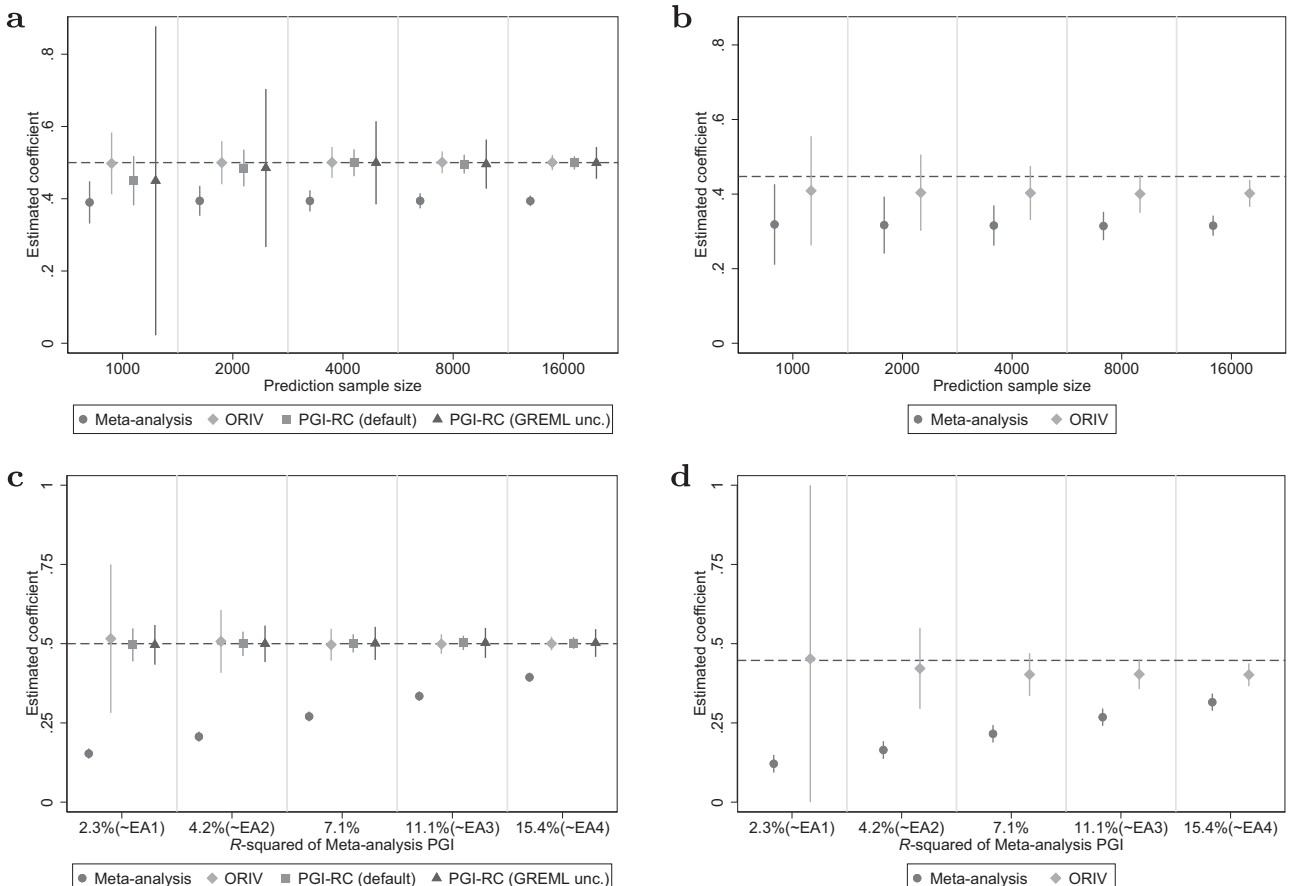

**Fig. 3 | Genetic nurture. a, c** Between-family analyses. **b, d** Within-family analyses. Data are presented as the estimated coefficients +/− 1.96 times the standard error (95% confidence interval) for the Polygenic Index (PGI) using meta-analysis (circles), Obviously Related Instrumental Variables (ORIV, rhombuses), the default PGI-RC procedure (squares), and the PGI-RC procedure, taking into account uncertainty in the GREML estimates (triangles). The top panels are for a scenario with genetic nurture but no assortative mating, and holding constant the GWAS discovery

sample such that the resulting meta-analysis PGI has an $R^2$ of 15.4%. The bottom panels are the same but now holding the discovery sample fixed at $N = 16,000$. The dashed line represents the true coefficient, which is equal to 0.5 (i.e., the square root of $h_{SNP}^2$) in the between-family design, and equal to the square root of the direct genetic effect $\sqrt{(h^2)} = \sqrt{(0.2)}$ in the within-family design. The simulation results are based on 100 replications.

contrast, the PGI-RC procedure is not sensitive at all since it re-scales a particular coefficient with the SNP-based heritability in the prediction sample. For a lower genetic correlation across the two discovery samples, the PGI-RC is completely insensitive, and ORIV is remarkably robust against deviations from a perfect genetic correlation between the two GWAS discovery samples. For a genetic correlation between the two GWAS samples of 0.75, ORIV is consistent, and even when the genetic correlation is as low as 0.5, the ORIV 95% confidence interval still includes the true coefficient.

**Summary.** The simulations suggest that it is virtually always beneficial to apply a measurement error correction compared to just using a meta-analysis based PGI. We find that both the PGI-RC as well as ORIV outperform the benchmark in all scenarios. Comparing among them, the PGI-RC is particularly valuable in between-family analyses when the GWAS discovery sample is very small (i.e., when the predictive power of a PGI is low) or when there exists an imperfect genetic correlation between the discovery and prediction sample (i.e., when the effects of SNPs on the outcome are very different across the discovery and prediction sample). In the presence of assortative mating, however, the PGI-RC is both more biased and less precise than ORIV. Moreover, the PGI-RC tends to be imprecise in case of small (i.e., $N < 1000$) prediction samples, and its confidence intervals are wider than those of ORIV when incorporating the uncertainty of the SNP-based heritability estimates. In those cases, as well as in within-family settings where the

PGI-RC correction is generally not available, ORIV seems the preferred alternative.

## Empirical illustration

In this section, we use OLS (using a meta-analysis based PGI), ORIV, and the PGI-RC to predict EA and height in a subsample of European ancestry siblings in the UK Biobank ($N = 35,282$). In the prediction sample, we first residualized the outcomes EA and height for sex, year of birth, month of birth, sex interacted with year of birth, and the first 40 principal components of the genetic relationship matrix. For both EA and height, we consider three PGIs: (i) a PGI based on the UKB discovery sample that excludes siblings and their relatives; (ii) a PGI based on the 23andMe, Inc. sample (EA) or the GIANT consortium (height; ref. [46]); and (iii) a PGI based on a meta-analysis of (i) and (ii). In addition, we construct two additional PGIs on the basis of randomly splitting the UKB discovery sample into two equal halves. All PGIs are constructed with the LDpred software[47] using as parametrization a default prior value of 1. We standardize the PGIs to have mean 0 and standard deviation 1 in the analysis sample. The standardization of the PGI has the advantage that the square of its estimated coefficient in a univariate regression is equal to the $R$-squared (see "Methods"). Additionally, by standardizing a given PGI we can interpret the resulting coefficient as a one standard deviation increase in the true latent PGI (i.e., the additive SNP factor). More details on the variables and their construction can be found in Supplementary Methods 2.

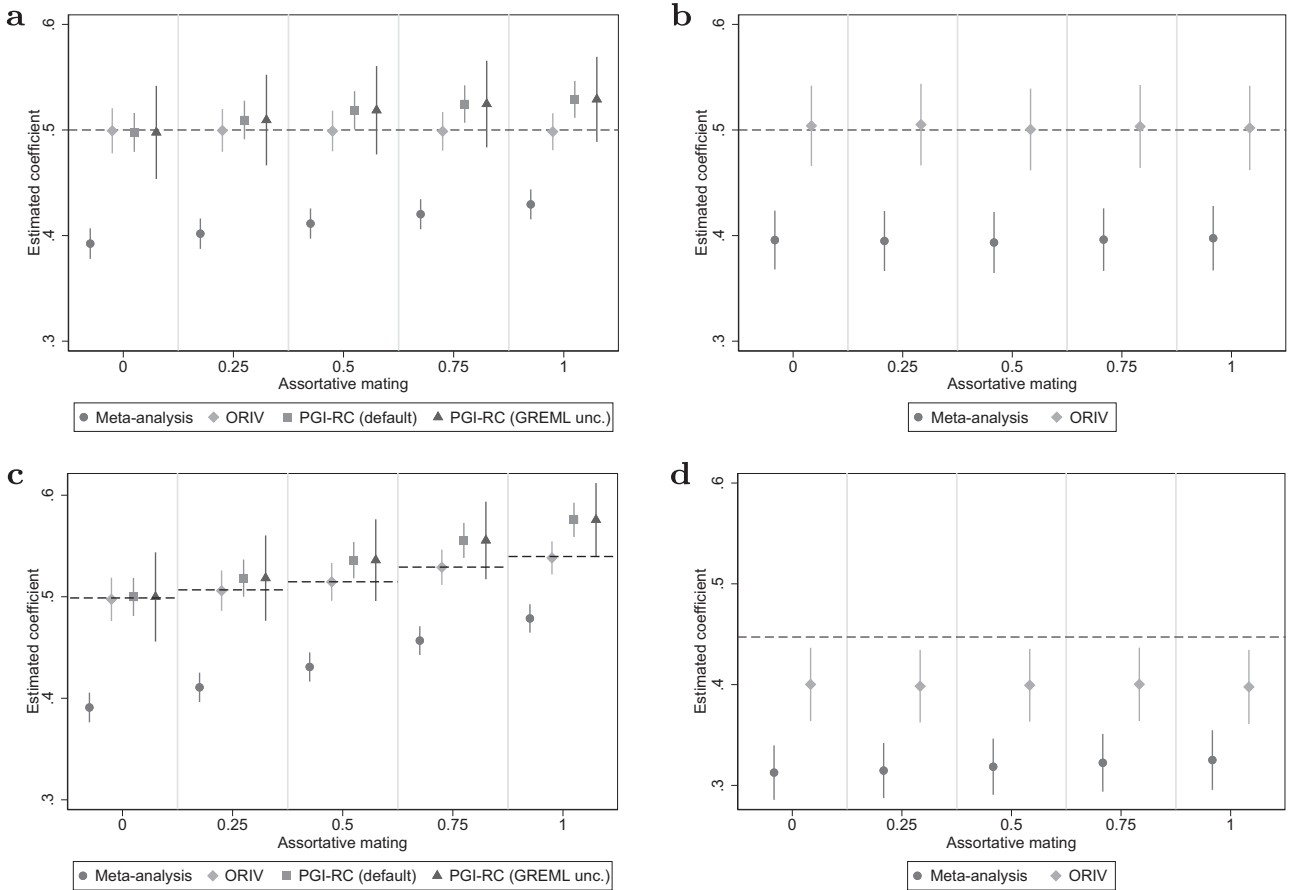

**Fig. 4 | Assortative mating. a**, **c** Between-family analyses. **b**, **d** Within-family analyses. Data are presented as the estimated coefficients +/− 1.96 times the standard error (95% confidence interval) for the Polygenic Index (PGI) using meta-analysis (circles), Obviously Related Instrumental Variables (ORIV, rhombuses), the default PGI-RC procedure (squares), and the PGI-RC procedure, taking into account uncertainty in the GREML estimates (triangles). The top two panels do not include genetic nurture while the bottom two panels model genetic nurture in addition to

AM. The simulations hold constant the Genome-wide Association Study (GWAS) discovery sample such that the resulting meta-analysis PGI has an $R^2$ of 15.4%, and the prediction sample size is held fixed at $N = 16{,}000$. The dashed line represents the true coefficient, which is equal to 0.5 (i.e., the square root of $h_{SNP}^2$) in the between-family analysis, and equal to the square root of the direct genetic effect $\sqrt{(h^2)} = \sqrt{(0.2)}$ in the within-family analysis. The simulation results are based on 100 replications.

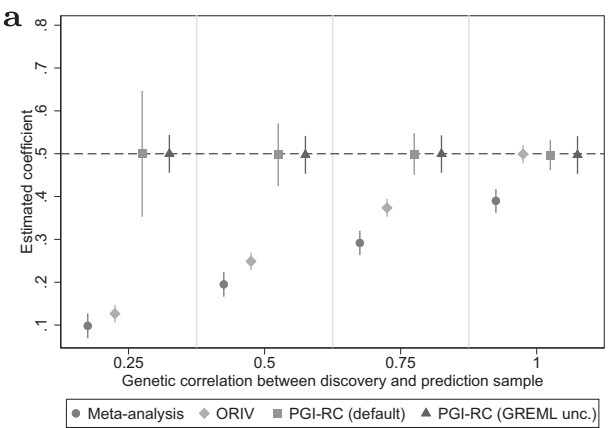
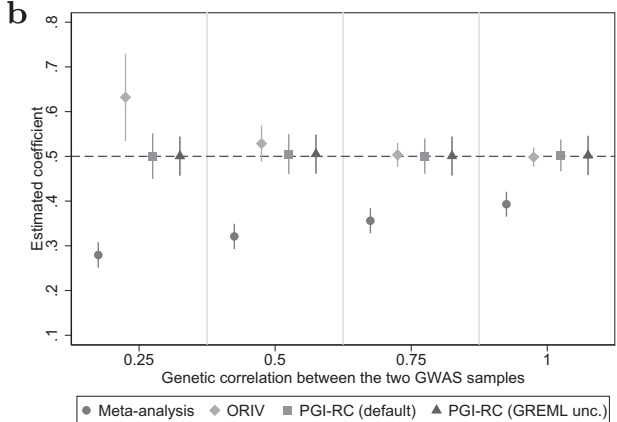

**Fig. 5 | Genetic correlation.** Data are presented as the estimated coefficients +/− 1.96 times the standard error (95% confidence interval) for the Polygenic Index (PGI) using meta-analysis (circles), Obviously Related Instrumental Variables (ORIV, rhombuses), the default PGI-RC procedure (squares), and the PGI-RC procedure, taking into account uncertainty in the GREML estimates (triangles) in a scenario without genetic nurture, without assortative mating, and varying levels of genetic

correlation between the GWAS and prediction samples (**a**) and between two GWAS samples that are meta-analyzed or used by ORIV (**b**). The simulations hold constant the Genome-wide Association Study (GWAS) discovery sample such that the resulting meta-analysis PGI has an $R^2$ of 15.4%, and the prediction sample size is held fixed at $N = 16{,}000$. Between-family analyses only. The dashed line represents the true coefficient. The simulation results are based on 100 replications.

**Table 2 | Results of the OLS and IV regressions explaining (residualized and standardized) educational attainment**

| | OLS (UKB) | OLS (23andMe) | OLS (Meta-analysis) | ORIV (2-sample) | ORIV (Split-sample) | PGI-RC (default) | PGI-RC (GREML unc.) |
|---|---|---|---|---|---|---|---|
| **Between-family results** | | | | | | | |
| Polygenic index | 0.258*** | 0.218*** | 0.276*** | 0.337*** | 0.323*** | 0.394*** | 0.394*** |
| | (0.005) | (0.005) | (0.005) | (0.007) | (0.007) | (0.007) | (0.024) |
| First-stage estimate | | | | 0.498*** | 0.489*** | | |
| | | | | (0.005) | (0.005) | | |
| First-stage $F$-statistic | | | | 11,919 | 11,061 | | |
| Incremental $R^2$ | 6.7% | 4.7% | 7.6% | | | | |
| Family fixed effects | NO | NO | NO | NO | NO | NO | NO |
| $N$ | 35,282 | 35,282 | 35,282 | 35,282 | 35,282 | 35,282 | 35,282 |
| **Within-family results** | | | | | | | |
| Polygenic index | 0.124*** | 0.115*** | 0.142*** | 0.184*** | 0.170*** | | |
| | (0.009) | (0.009) | (0.009) | (0.012) | (0.013) | | |
| First-stage estimate | | | | 0.460*** | 0.436*** | | |
| | | | | (0.006) | (0.006) | | |
| First-stage $F$-statistic | | | | 6068 | 5158 | | |
| Incremental $R^2$ | 1.5% | 1.3% | 2.0% | | | | |
| Family fixed effects | YES | YES | YES | YES | YES | | |
| $N$ | 35,282 | 35,282 | 35,282 | 35,282 | 35,282 | | |

Notes: *$p$ value < 0.10; **$p$ value < 0.05; ***$p$ value < 0.01 of a two-sided $t$ test (OLS, PGI-RC) or two-sided $z$-test (ORIV) without adjustments for multiple comparisons. In all regressions the dependent variable is residualized educational attainment (EA, standardized to have mean 0 and standard deviation 1), where the residuals are obtained from a regression of EA on sex, year of birth, month of birth, sex interacted with year of birth, and the first 40 principal components of the genetic relationship matrix. Standard errors are robust and clustered at the family level, and in case of ORIV also at the individual level. OLS (UKB) refers to the model with the PGI constructed using the UKB non-sibling (i.e., excluding all siblings and their relatives) sample. OLS (23andMe) refers to the model with the PGI constructed using the 23andMe summary statistics. OLS (Meta-analysis) uses a PGI constructed using a meta-analysis of GWAS summary statistics of the UKB non-sibling sample and the 23andMe sample. ORIV (2-sample) refers to a 2SLS estimation using the PGIs from the UKB non-sibling sample and 23andMe as instrumental variables for each other. ORIV (Split-sample) refers to a 2SLS estimation where the summary statistics derive from a random split of the UKB sample. PGI-RC refers to the PGI repository correction, where (default) refers to the conventional application of the method, whereas (GREML unc.) refers to the case where we incorporate the uncertainty in the estimation of the SNP-based heritability.

**SNP-based heritability.** Using LDSC and GREML, we estimate the SNP-based heritability of EA in the prediction sample to be 0.162 (s.e. 0.027) and 0.155 (s.e. 0.019), respectively, which is somewhat lower than estimates from other samples in the literature. For height, the SNP-based heritability is estimated to be 0.497 (s.e. 0.039) using LDSC, and 0.530 (s.e. 0.020) using GREML. These results were obtained using one randomly selected sibling per family in the sibling subsample ($N = 18,989$ for EA, $N = 18,913$ for height). To obtain the LDSC estimate, the GWAS summary statistics were computed using FastGWA[48]. In the GREML analysis[49], the analysis sample was slightly lower ($N = 17,696$ for EA, $N = 17,849$ for height) because we excluded closely related individuals using the default relatedness cut-off of 0.025. These SNP-based heritabilities are a useful benchmark, as they constitute an upper bound on the $R^2$ we can achieve in our sample using a PGI[9]. The SNP-based heritabilities additionally are a crucial input for disattenuating the OLS estimator in the PGI-RC[6]. In fact, in the univariate case, the SNP-based heritability obtained using GREML is actually the PGI-RC estimate.

**Educational attainment (EA).** Table 2 shows the results of regressions of residualized years of education (EA) (standardized to have mean 0 and standard deviation 1 in the sample) on the various PGIs. Figure 6a visualizes the results in terms of estimated heritability (i.e., the square of the standardized PGI coefficient). Meta-analyzing summary statistics from independent samples increases the standardized effect size and associated predictive power of the PGI compared with using the individual PGIs. That is, the PGI based on the meta-analysis of the UKB sample (excluding siblings and their relatives) and 23andMe delivers a standardized effect size of 0.276 (Column 3), implying an estimated heritability of 7.6%. This estimate is clearly higher than the effect sizes and heritability estimates obtained when using the UKB or 23andMe samples on their own (Columns 1 and 2; and the first two estimates of Fig. 6a). Nevertheless, the meta-analysis PGI still delivers an $R^2$ that is

substantially below the estimates of the SNP-based heritability of 15.5–16.0%.

Column 4 of Table 2 shows the ORIV estimates employing the PGIs obtained from UKB and 23andMe as instrumental variables for each other. The ORIV standardized effect estimate is 0.337. Whereas the $R^2$ of an IV regression is not meaningful[50], we can estimate the implied heritability $\hat{h}^2_{SNP}$ by squaring the coefficient, which gives 11.4%. While, unlike in the simulations, we do not know the true standardized effect size, it is reassuring that the implied heritability estimate is close to our empirical estimates of the SNP-based heritability of 15.5% (s.e. 1.9%; GREML) and 16.2% (s.e. 2.7%; LDSC), with the confidence intervals overlapping. In Column 5, we additionally present the ORIV results based on two PGIs that were constructed using two random halves of the UKB discovery sample. The IV assumptions are more likely to hold in this scenario since the samples are equally sized and they originate from the exact same environmental context. In particular, we estimate the genetic correlation between the 23andMe and UKB summary statistics to be 0.878 (s.e. 0.011), whereas the genetic correlation between the split-sample UKB summary statistics is 1.000 (s.e. <0.001). The resulting coefficient and implied heritability of 10.4% of the split-sample ORIV are only slightly below the two-sample ORIV results in Column 4, and considerably higher than the estimate obtained with the meta-analysis PGI.

When comparing ORIV to the PGI-RC, which in this univariate context equals the GREML estimate, we observe that the ORIV estimators are somewhat below the PGI-RC estimates (i.e., ORIV estimators may exhibit some bias). At the same time, when accounting for the uncertainty in the GREML estimate of the heritability $h^2_{SNP}$ that underlies the PGI-RC (column 7 of Table 2 and bottom row of Fig. 6a), the precision of ORIV estimators is considerably higher (i.e., ORIV estimators have lower variance). Hence, there is a bias-variance trade-off between ORIV and the PGI-RC in this context.

a

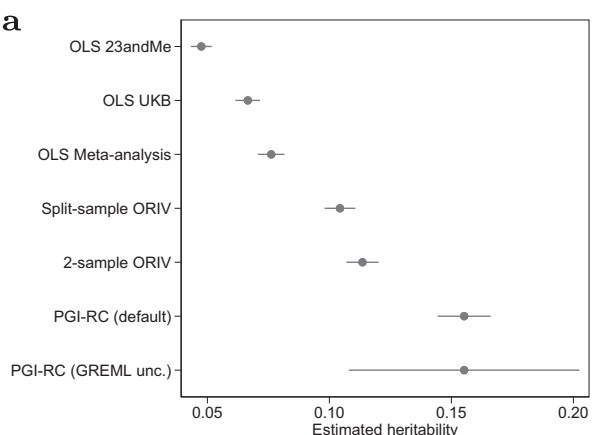

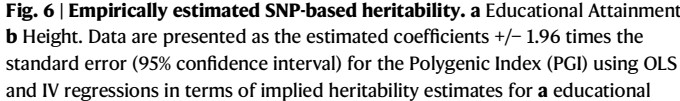

b

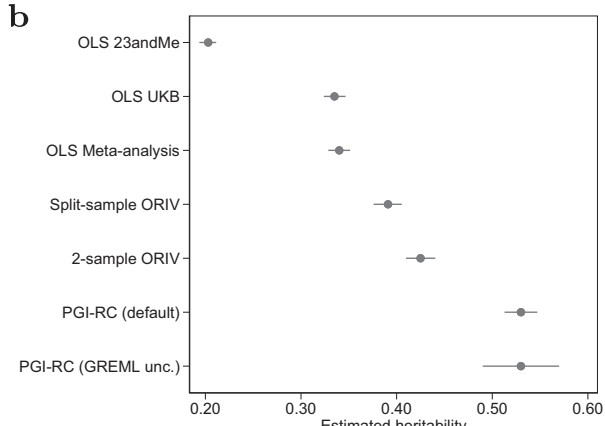

**Fig. 6 | Empirically estimated SNP-based heritability. a** Educational Attainment. **b** Height. Data are presented as the estimated coefficients +/− 1.96 times the standard error (95% confidence interval) for the Polygenic Index (PGI) using OLS and IV regressions in terms of implied heritability estimates for **a** educational attainment (EA) and **b** height. $n = 35{,}282$ independent individuals. The implied heritability is computed on the basis of the square of the standardized coefficients (see Eq. (24)), and its standard error is obtained using the Delta method. Rest as in Table 2.

The results in the bottom panel of Table 2 are obtained using regressions that include family fixed effects. This approach only relies on within-family variation in the PGIs and therefore uncovers direct genetic effects[51]. It bears repeating that in this context the PGI-RC cannot be applied, and so the relevant comparison here is between regular OLS and ORIV. The standardized effect estimates are substantially smaller within-families than they are between-families. This finding reflects an upward bias in the between-family estimates as a result of population phenomena, most notably genetic nurture[15,51,52]. More specifically, in line with the literature[18,53,54], our within-family ORIV estimates are around 45% smaller than the between-family ORIV estimates. As in the between-family analyses, applying ORIV within-families increases the coefficient of the PGI compared to a meta-analysis or using standalone individual PGIs. ORIV estimates the standardized direct genetic effect to be 0.184 (using PGI from two samples −UKB and 23andMe) and 0.170 (using split-sample UKB PGIs). This estimate may still be prone to attenuation bias since the PGIs are based on a GWAS that did not consider indirect genetic effects from relatives (see also "Methods")[55,56], but this estimate does represent a tighter lower bound on the direct genetic effect.

**Height**. Table 3 and Fig. 6b present the results of the regressions with height as the outcome variable. The standardized effect sizes for height are considerably larger than for EA, consistent with the higher heritability of height. For example, a meta-analysis PGI based upon the UKB and the GIANT consortium GWAS summary statistics reaches a standardized effect size of 0.583, which corresponds to an incremental $R$-squared of 34%. It is also noteworthy that for height the between- and within-family results do not differ as much as they do for EA. Again, this is in line with the literature[18] which generally finds genetic nurture to be more important for behavioral outcomes such as EA than for anthropometric outcomes like height.

Despite the differences in heritability and in the role of genetic nurture, we reach similar conclusions for height as for EA in the comparison of OLS (meta-analysis), ORIV, and the PGI-RC. The two-sample ORIV estimation is 25% (between-family) and even 30% (within-family) higher when using ORIV compared to a meta-analysis PGI. The confidence interval for the PGI-RC incorporating uncertainty in the GREML estimate is again larger than the confidence interval of the ORIV estimate, yet the point estimate of the PGI-RC is significantly higher than the one for ORIV. In the within-family analysis, ORIV delivers the tightest lower bound on the direct genetic effect for height, which is estimated to be above 0.6. With height being a typical trait to

test new quantitative genetics methodologies[49], these empirical findings build confidence that our conclusions from the simulations apply more broadly.

**Assortative mating**. As shown in the simulations, depending on the degree of assortative mating, the SNP-based heritability may be overestimated, and hence the PGI-RC could be overestimating the true effect size more severely than ORIV. With EA being a trait that exhibits a considerable degree of assortative mating[38,39,57], it is not entirely clear whether the estimated SNP-based heritability of 15.5% (and corresponding standardized effect of ~0.4) is the correct target, or that it is overestimating the true effect. Similarly, estimates of assortative mating for height are in the order of 0.23[58], again leaving potential for bias in the GREML estimates that underlie the PGI-RC. For this reason, in Supplementary Results 3, we compare the relative performance of ORIV and the PGI-RC for a trait with a similar level of heritability but that exhibits considerably less assortative mating: diastolic blood pressure (DBP)[59]. In line with the absence of assortative mating, the between-family and within-family results are highly concordant in case of DBP. Similar to the results for EA and height, ORIV increases the estimated coefficient by 27% compared to an OLS regression on basis of the meta-analyzed PGI. Again similar to EA and height, the confidence interval around the PGI-RC estimates is wider, but ORIV tends to produce slightly lower point estimates than the PGI-RC. In sum, it seems that our empirical findings are not driven by assortative mating in EA or height.

## Discussion

The increasing availability of genetic data over the last decade has stimulated genetic discovery in GWAS studies and has led to increases in the predictive power of polygenic indices (PGIs). Phenotypes such as educational attainment (EA) and height are currently at a critical turning point at which boosting the GWAS sample size further will only increase the predictive power of the PGIs at a marginal and diminishing rate. As a result, for the foreseeable future, regressions involving PGIs will be subject to an attenuation bias resulting from measurement error in the estimated GWAS coefficients. In this study, we compared two approaches that attempt to overcome attenuation bias in the estimated regression coefficients of a PGI: Obviously-Related Instrumental Variables (ORIV) and the PGI Repository Correction (PGI-RC).

Extensive simulations show that in between-family analyses, the comparison of ORIV versus the PGI-RC is subject to a bias-variance

**Table 3 | Results of the OLS and IV regressions explaining (residualized and standardized) height**

| | OLS (UKB) | OLS (GIANT) | OLS (Meta-analysis) | ORIV (2-sample) | ORIV (Split-sample) | PGI-RC (default) | PGI-RC (GREML unc.) |
|---|---|---|---|---|---|---|---|
| **Between-family results** | | | | | | | |
| Polygenic index | 0.579*** | 0.450*** | 0.583*** | 0.652*** | 0.625*** | 0.728*** | 0.728*** |
| | (0.005) | (0.005) | (0.005) | (0.006) | (0.006) | (0.006) | (0.014) |
| First-stage estimate | | | | 0.622*** | 0.586*** | | |
| | | | | (0.004) | (0.004) | | |
| First-stage $F$-statistic | | | | 27,261 | 20,792 | | |
| Incremental $R^2$ | 33.5% | 20.3% | 34.0% | | | | |
| Family fixed effects | NO | NO | NO | NO | NO | NO | NO |
| $N$ | 35,282 | 35,282 | 35,282 | 35,282 | 35,282 | 35,282 | 35,282 |
| **Within-family results** | | | | | | | |
| Polygenic index | 0.521*** | 0.415*** | 0.537*** | 0.614*** | 0.571*** | | |
| | (0.007) | (0.007) | (0.007) | (0.008) | (0.009) | | |
| First-stage estimate | | | | 0.600*** | 0.558*** | | |
| | | | | (0.005) | (0.005) | | |
| First-stage $F$-statistic | | | | 15,735 | 11,150 | | |
| Incremental $R^2$ | 27.1% | 17.2% | 28.9% | | | | |
| Family fixed effects | YES | YES | YES | YES | YES | | |
| $N$ | 35,282 | 35,282 | 35,282 | 35,282 | 35,282 | | |

Notes: *$p$ value < 0.10; **$p$ value < 0.05; ***$p$ value < 0.01 of a two-sided $t$ test (OLS, PGI-RC) or two-sided $z$-test (ORIV) without adjustments for multiple comparisons. In all regressions the dependent variable is residualized height (standardized to have mean 0 and standard deviation 1), where the residuals are obtained after a regression of height on controlling for sex, year of birth, month of birth, sex interacted with year of birth, and the first 40 principal components of the genetic relationship matrix. Standard errors are robust and clustered at the family level, and in case of ORIV also at the individual level. OLS (UKB) refers to the model with the PGI constructed using the UKB non-sibling (i.e., excluding all siblings and their relatives) sample. OLS (GIANT) refers to the model with the PGI constructed using the GIANT summary statistics. OLS (Meta-analysis) uses a PGI constructed using a meta-analysis of GWAS summary statistics of the UKB non-sibling sample and the GIANT sample. ORIV (2-sample) refers to a 2SLS estimation using the PGIs from the UKB non-sibling sample and Giant as instrumental variables for each other. ORIV (Split-sample) refers to a 2SLS estimation where the summary statistics derive from a random split of the UKB sample. PGI-RC refers to the PGI repository correction, where (default) refers to the conventional application of the method, whereas (GREML unc.) refers to the case where we incorporate the uncertainty in the estimation of the SNP-based heritability.

trade-off that will differ across applications. The PGI-RC performs especially well (in terms of Root Mean Squared Error) compared with ORIV when there exists imperfect genetic correlation between the discovery and prediction sample, or when the GWAS discovery sample is relatively small. These conclusions hold even when incorporating the additional uncertainty induced by first estimating the SNP-based heritability. However, when a sizable discovery sample is available that has a near-perfect genetic correlation with the prediction sample, ORIV tends to perform better than the PGI-RC. Moreover, when there exists considerable assortative mating on the basis of the outcome variable, ORIV also tends to perform significantly better compared with the PGI-RC. In within-family analyses, ORIV is the most convenient way of estimating direct genetic effects. The simulations suggest that applying ORIV within-families provides consistent estimates in the absence of genetic nurture and assortative mating, and tightens the lower bound on the direct genetic effects in the presence of genetic nurture and assortative mating.

We empirically tested these predictions using UK Biobank data on educational attainment (EA) and height and largely confirm the simulation results. Both ORIV and the PGI-RC outperform a meta-analysis based PGI in terms of bias and RMSE. Among them, in our application, ORIV tends to underestimate the SNP-heritability somewhat (11% versus 15% for EA, and 43% versus 53% for height), but tends to have smaller standard errors than the PGI-RC, especially when the PGI-RC incorporates the uncertainty in the GREML estimates of the SNP-based heritability. On the basis of within-family analyses, in our application, ORIV estimated the standardized direct genetic effect to be around 0.18 for EA and 0.6 for height, respectively a 30% (EA) and 14% (height) increase compared with a meta-analysis PGI. Similar findings for diastolic blood pressure provide reassurance that assortative mating on EA and height was not driving these empirical findings.

In sum, the application of the PGI-RC or ORIV in empirical applications requires a careful assessment of the setting in which the correction is applied. From a practical point of view, it is a particular advantage of ORIV that it can be easily implemented in all standard statistical software packages (see https://github.com/geighei/ORIV for implementations in STATA and R). Moreover, ORIV does not require external information on the SNP-based heritability. This feature makes ORIV particularly attractive for within-family studies aiming to estimate direct genetic effects, because SNP-based heritability estimates do not solely capture direct genetic effects, but also incorporate indirect genetic effects from relatives, assortative mating, and population stratification[60]. Whereas Young et al.[37] and Eilertsen et al.[61] developed approaches for separating direct and indirect genetic effects, both methods require genetic data from unrelated individuals and both of their parents, samples of which are currently very rare. Hence, ORIV is more flexible since it does not require external information on the "true" level of disattenuation, which is hard to obtain in within-family settings or for traits that are subject to considerable assortative mating.

There are alternative approaches to deal with measurement error than those analyzed in this study. Simulation-extrapolation (SIMEX)[62,63] is an approach that also exploits external information on the SNP-based heritability, somewhat similar to the PGI-RC[6], yet relying on simulations. The advantage of ORIV over SIMEX is that it does not require external information or simulations, and can also be applied within-families. Notable other techniques to deal with measurement error are the Generalized Method of Moments (GMM[64]) and Structural Equation Modeling (SEM[28,65]). Since IV can be seen as a special case of GMM and SEM models[28,66], the differences between the approaches are typically negligible in linear models. The distributional assumptions are somewhat stronger in SEM compared with IV, and including family fixed effects tends to be

cumbersome[67–69]. A possible advantage of SEM is however its flexibility in allowing the factor loadings of the two individual PGIs to be different. This could be especially relevant when the sample sizes and/or genetic correlation with the prediction sample differ substantially across two GWAS discovery samples. An extensive comparison of ORIV versus (genomic) SEM or GMM is beyond the scope of this paper, but we anticipate that differences will typically be small unless the precision of the two independent PGIs differs substantially.

Whereas we have shown that both ORIV as well as the PGI-RC are superior to a regular meta-analysis PGI in terms of overcoming attenuation bias, this does not mean that further collection of additional genotyped samples is useless. In contrast, larger sample sizes are essential in identifying specific genetic variants that affect the phenotype of interest, allowing one to investigate the biological mechanisms driving these effects. Further, whereas the PGI-RC and ORIV are useful "scaling tools" to estimate the regression coefficient of the latent "true" PGI, they do not boost the predictive power of a PGI in out-of-sample applications. Finally, applying measurement error corrections using ORIV or the PGI-RC are not a substitute for within-family GWASs. The collection of family samples is the only way to explicitly control for the indirect genetic effects from relatives that plague the interpretation of the effects of PGIs in between-family studies. The collection of genetic data of family samples is on the rise[30], but their sample sizes are still comparatively small. The results of the present study suggest that the application of ORIV could help to reduce attenuation bias in regression coefficients for PGIs constructed using the results of within-family GWASs.

## Methods

### Conceptual model

Consider a simple linear fixed effects model in which a dependent variable $Y$ (e.g., educational attainment) is influenced by many genetic variants:

$$Y = \alpha + \sum_{j=1}^{J} \beta_j^{GWAS} \mathrm{SNP}_j + \varepsilon \qquad (3)$$

where $J$ is the number of genetic variants (single-nucleotide polymorphisms, SNPs) included, $SNP_j$ represents the number of effect alleles an individual possesses at locus $j$, and $\beta_j^{GWAS}$ is the coefficient of SNP $j$. The true data generating process would also include the effects of maternal and paternal SNPs, because only conditional on parental genotypes the variation in SNPs is random and hence exogenous. We discuss the use of family data briefly below, but ignore the effects of parental genotype as well as environmental factors in the following discussion for simplicity.

The dependent variable $Y$ is assumed to be standardized with mean zero and standard deviation 1 ($\sigma_Y = 1$). The true latent polygenic index PGI* is then defined as:

$$\mathrm{PGI}^* = \sum_{j=1}^{J} \beta_j^{GWAS} \mathrm{SNP}_j. \qquad (4)$$

If we would observe the true polygenic index PGI*, then running the OLS regression

$$Y = \alpha + \beta \mathrm{PGI}^* + \varepsilon \qquad (5)$$

yields

$$\hat{\beta} = \frac{\mathrm{Cov}(Y, \mathrm{PGI}^*)}{\mathrm{V}(\mathrm{PGI}^*)} = \frac{\mathrm{Cov}(\alpha + \beta \mathrm{PGI}^* + \varepsilon, \mathrm{PGI}^*)}{\sigma_{\mathrm{PGI}^*}^2} = \beta \frac{\sigma_{\mathrm{PGI}^*}^2}{\sigma_{\mathrm{PGI}^*}^2} = \beta$$

where $\beta$ measures what happens to the outcome $Y$ when the true latent PGI* increases with 1 unit in the analysis sample. Since a 1 unit increase in the PGI* is not straightforward to interpret, researchers are typically more interested in $\beta \times \sigma_{\mathrm{PGI}^*}$, i.e., a one standard deviation increase in the true PGI. This estimate can be obtained by standardizing the PGI:

$$\mathrm{PGI}_{st}^* = \frac{\mathrm{PGI}^* - \mu_{\mathrm{PGI}^*}}{\sigma_{\mathrm{PGI}^*}},$$

where $\mu_{\mathrm{PGI}^*}$ is the mean of the true PGI, and $\sigma_{\mathrm{PGI}^*}$ is the standard deviation of the true PGI. If we now run the regression $Y = \alpha + \beta_{st} \mathrm{PGI}_{st}^* + \varepsilon$, then the resulting estimator is:

$$\hat{\beta}_{st} = \frac{\mathrm{Cov}(Y, \mathrm{PGI}_{st}^*)}{\mathrm{V}(\mathrm{PGI}_{st}^*)} = \mathrm{Cov}\left(\alpha + \beta \mathrm{PGI}^* + \varepsilon, \frac{\mathrm{PGI}^* - \mu_{\mathrm{PGI}^*}}{\sigma_{\mathrm{PGI}^*}}\right)$$
$$= \beta \frac{\sigma_{\mathrm{PGI}^*}^2}{\sigma_{\mathrm{PGI}^*}} = \beta \sigma_{\mathrm{PGI}^*} \equiv \beta_{st}. \qquad (6)$$

Apart from an arguably easier interpretation, the standardization of the PGI has the added advantage that there is a close connection between the estimated coefficient and the $R$-squared of this univariate regression. That is, in a univariate regression, the $R$-squared measures the squared correlation between the outcome and the independent variable, and it can be compared to the upper bound represented by the SNP-based heritability.

### Measurement error in the polygenic index

In practice, any estimated PGI is a proxy for the true latent polygenic index PGI* because it is measured with error:

$$\mathrm{PGI} = \mathrm{PGI}^* + \nu, \quad \nu \sim N(0, \sigma_\nu^2)$$

where we assume that the measurement error $\nu$ is classical in the sense that it is uncorrelated to the error term in Eq. (3). If we estimate the regression $Y = \alpha + \beta \mathrm{PGI} + \varepsilon$, then measurement error in the PGI attenuates the coefficient of the PGI on $Y$:

$$\hat{\beta} = \frac{\mathrm{Cov}(Y, \mathrm{PGI})}{\mathrm{V}(\mathrm{PGI})} = \frac{\mathrm{Cov}(\alpha + \beta \mathrm{PGI}^* + \varepsilon, \mathrm{PGI}^* + \nu)}{\mathrm{V}(\mathrm{PGI}^* + \nu)} = \beta \frac{\sigma_{\mathrm{PGI}^*}^2}{\sigma_{\mathrm{PGI}^*}^2 + \sigma_\nu^2} < \beta. \qquad (7)$$

If—as is common in the literature—the observed PGI is standardized to obtain $\mathrm{PGI}_{st}$, it follows that:

$$\mathrm{PGI}_{st} = \frac{\mathrm{PGI} - \mu_{\mathrm{PGI}}}{\sigma_{\mathrm{PGI}}} = \frac{\mathrm{PGI}}{\sqrt{\sigma_{\mathrm{PGI}^*}^2 + \sigma_\nu^2}} = \frac{\mathrm{PGI}^* + \nu}{\sqrt{\sigma_{\mathrm{PGI}^*}^2 + \sigma_\nu^2}}. \qquad (8)$$

The resulting standardized coefficient of $\mathrm{PGI}_{st}$ on $Y$ is given by

$$\hat{\beta}_{st} = \frac{\mathrm{Cov}(Y, \mathrm{PGI}_{st})}{\mathrm{V}(\mathrm{PGI}_{st})} = \frac{\mathrm{Cov}\left(\alpha + \beta \mathrm{PGI}^* + \varepsilon, \frac{\mathrm{PGI}^* + \nu}{\sqrt{\sigma_{\mathrm{PGI}^*}^2 + \sigma_\nu^2}}\right)}{1}$$
$$= \beta \frac{\sigma_{\mathrm{PGI}^*}^2}{\sqrt{\sigma_{\mathrm{PGI}^*}^2 + \sigma_\nu^2}} = \beta_{st} \frac{\sigma_{\mathrm{PGI}^*}}{\sqrt{\sigma_{\mathrm{PGI}^*}^2 + \sigma_\nu^2}} < \beta_{st}. \qquad (9)$$

Note that standardizing the observed PGI with respect to its own standard deviation is a combination of standardizing with respect to the standard deviation of the true PGI as well as the standard deviation of measurement error (see also ref. 28). Therefore, Eq. (9) shows that the estimate should be interpreted as the effect of a 1 standard deviation increase in the observed PGI (and not the true latent PGI). Hence, this estimate does not just underestimate the true $\beta$ coefficient

due to measurement error but should also be interpreted on a different scale than the effect of the true PGI[6,28,70].

Equation (9) can be rewritten as:

$$\frac{\sigma^2_{PGI^*}}{\sigma^2_{PGI^*} + \sigma^2_\nu} = \frac{\hat{\beta}^2_{st}}{\beta^2_{st}} = \frac{R^2}{h^2_{SNP}} \qquad (10)$$

where we have used that the square of the standardized coefficient provides an estimate of the heritability. Hence, Eq. (10) shows that the bias in OLS is determined by the ratio of the estimated $R$-squared over the SNP-based heritability.

## PGI repository correction (PGI-RC)

Becker et al.[6] exploit their equivalent of Eq. (10) to directly derive an estimator that corrects for measurement error. The authors invert Eq. (10) and take the square root to obtain a disattenuation factor:

$$\sqrt{\frac{\beta^2_{st}}{\hat{\beta}^2_{st}}} = \sqrt{\frac{h^2_{SNP}}{R^2}} = \frac{\sqrt{\sigma^2_{PGI^*} + \sigma^2_\nu}}{\sigma_{PGI^*}}. \qquad (11)$$

In turn, they suggest—in the univariate case—to multiply the estimated coefficient $\hat{\beta}_{st}$ of the standardized PGI by the disattenuation factor to obtain effects of the true PGI (they call this the "the additive SNP factor") free from measurement error:

$$\hat{\beta}_{st}\sqrt{\frac{h^2_{SNP}}{R^2}} = \beta_{st}\frac{\sigma_{PGI^*}}{\sqrt{\sigma^2_{PGI^*} + \sigma^2_\nu}}\frac{\sqrt{\sigma^2_{PGI^*} + \sigma^2_\nu}}{\sigma_{PGI^*}} \qquad (12)$$

$$= \beta_{st}.$$

This procedure conveniently ensures that the estimated effect of the PGI is equal to the estimated SNP-based heritability, which the developers suggest to obtain using GREML. Hence, the bias of the product estimator in Eq. (12) is zero if the GREML assumptions are met and the researcher can compute the SNP-based heritability in the same sample as where the analysis is conducted.

The standard error of the PGI-RC estimator is discussed in the Supplementary Information of Becker et al.[6]. When treating the factor $\sqrt{h^2_{SNP}/R^2}$ as a fixed non-stochastic scaling term, the variance can be derived as:

$$V\left(\hat{\beta}_{st}\sqrt{\frac{h^2_{SNP}}{R^2}}\right) = \sqrt{\frac{h^2_{SNP}}{R^2}}\text{Var}\left(\hat{\beta}_{st}\right)\sqrt{\frac{h^2_{SNP}}{R^2}} \qquad (13)$$

$$= \frac{h^2_{SNP}}{R^2}V\left(\hat{\beta}_{st}\right)$$

and the resulting standard error is then simply given by

$$\text{s.e.}\left(\hat{\beta}_{st}\sqrt{\frac{h^2_{SNP}}{R^2}}\right) = \sqrt{\frac{h^2_{SNP}}{R^2}}\text{s.e.}\left(\hat{\beta}_{st}\right) \qquad (14)$$

That is, both the coefficient as well as the standard error of the original standardized coefficient are simply scaled by the same factor $\sqrt{h^2_{SNP}/R^2}$. This standard error is treated as the default and also implemented in the accompanying software.

However, as acknowledged in the Supplementary Information (Section 5, pages 14 and 15) of Becker et al.[6], in practice the scaling factor is not a fixed immutable statistic, but rather a stochastic factor that is estimated in a first step. Hence, the true standard error of the

PGI-RC equals the standard error of the square root of the SNP-based heritability obtained through GREML. Using the Delta method, this leads to a standard error

$$\text{s.e.}\left(\hat{\beta}_{st}\sqrt{\frac{\hat{h}^2_{SNP}}{\hat{R}^2}}\right) = \text{s.e.}\left(\hat{h}_{SNP}\right) = \frac{\text{s.e.}\left(\hat{h}^2_{SNP}\right)}{2\hat{h}_{SNP}} \qquad (15)$$

In our simulations as well as empirical applications, we report both standard errors (Eqs. (14) and (15)) to allow for an accurate comparison across methods. In rare cases, in a very small prediction sample, the estimated SNP heritability in our simulations is estimated to be very small. This results in a huge standard error as can be seen in Eq. (15). To avoid that these rare cases have a large influence on our mean comparisons, we ignore runs that produce a SNP-based heritability lower than 0.01.

## Instrumental variables

An alternative way of addressing measurement error is instrumental variables (IV regression). It has long been recognized in the econometrics literature[71,72] that when at least two independent measures of the same construct (independent variable) are available, it is possible to retrieve a consistent effect of this construct on an outcome through IV estimation.

In terms of formulas, if we have two measures for the true PGI*, $\text{PGI}_1 = \text{PGI}^* + \nu_1$ and $\text{PGI}_2 = \text{PGI}^* + \nu_2$, with $\text{Cov}(\nu_1, \nu_2) = 0$ and $\frac{\sigma^2_{\nu_1}}{\sigma^2_{PGI_1}} = \frac{\sigma^2_{\nu_2}}{\sigma^2_{PGI_2}} = \frac{\sigma^2_\nu}{\sigma^2_{PGI}}$. Then:

$$\text{Cov}(\text{PGI}_1, \text{PGI}_2) = \text{Cov}(\text{PGI}^* + \nu_1, \text{PGI}^* + \nu_2) = \text{Cov}(\text{PGI}^*, \text{PGI}^*) = \sigma^2_{PGI^*}; \qquad (16)$$

$$\text{Corr}(\text{PGI}_1, \text{PGI}_2) = \frac{\text{Cov}(\text{PGI}_1, \text{PGI}_2)}{\sigma_{PGI_1}\sigma_{PGI_2}} = \frac{\sigma^2_{PGI^*}}{\sigma^2_{PGI^*} + \sigma^2_\nu}. \qquad (17)$$

Hence, the correlation between the two PGIs can be used to correct for the attenuation bias that plagues the interpretation of the OLS estimates in Eq. (7). More formally, if we use $\text{PGI}_2$ as an instrumental variable (IV) for $\text{PGI}_1$, then the IV estimator is the ratio of the reduced form (regression of $Y$ on $\text{PGI}_2$) and the first stage (regression on $\text{PGI}_1$ on $\text{PGI}_2$):

$$\hat{\beta}^{IV} = \frac{\frac{\text{Cov}(Y, \text{PGI}_2)}{\text{V}(\text{PGI}_2)}}{\frac{\text{Cov}(\text{PGI}_1, \text{PGI}_2)}{\text{V}(\text{PGI}_2)}} = \frac{\text{Cov}\left(\alpha + \beta\text{PGI}^* + \varepsilon, \text{PGI}^* + \nu_2\right)}{\text{Cov}\left(\text{PGI}^* + \nu_1, \text{PGI}^* + \nu_2\right)} \qquad (18)$$

$$= \beta\frac{\sigma^2_{PGI^*}}{\sigma^2_{PGI^*}} = \beta. \qquad (19)$$

As a consequence, IV regression is able to estimate the unstandardized coefficient of the true latent PGI in a consistent way. But since unstandardized coefficients are hard to interpret, one is typically more interested in the standardized coefficient. In this case, the IV estimator is given by:

$$\hat{\beta}^{IV}_{st} = \frac{\frac{\text{Cov}(Y, \text{PGI}_{2,st})}{\text{V}(\text{PGI}_{2,st})}}{\frac{\text{Cov}(\text{PGI}_{1,st}, \text{PGI}_{2,st})}{\text{V}(\text{PGI}_{2,st})}} = \frac{\text{Cov}\left(\alpha + \beta\text{PGI}^* + \varepsilon, \frac{\text{PGI}^* + \nu_2}{\sqrt{\sigma^2_{PGI^*} + \sigma^2_\nu}}\right)}{\text{Cov}\left(\frac{\text{PGI}^* + \nu_1}{\sqrt{\sigma^2_{PGI^*} + \sigma^2_\nu}}, \frac{\text{PGI}^* + \nu_2}{\sqrt{\sigma^2_{PGI^*} + \sigma^2_\nu}}\right)} \qquad (20)$$

$$= \beta\sqrt{\sigma^2_{PGI^*} + \sigma^2_\nu} \geq \beta\sigma_{PGI^*}.$$

Importantly, the standardized IV estimator is not equal to the effect of a one standard deviation increase in the true PGI since the PGI is

standardized with respect to the standard deviation of the observed instead of the true latent PGI. As a result, IV overestimates the true standardized coefficient. However, a way to retrieve the effect of a 1 standard deviation increase in the true latent PGI would be to scale the standardized IV coefficient:

$$\frac{\sigma_{PGI^*}}{\sqrt{\sigma_{PGI^*}^2 + \sigma_\nu^2}} \hat{\beta}_{st}^{IV} = \beta\sigma_{PGI^*}. \tag{21}$$

Although the scaling factor is unobserved, an estimate is given by the square root of the correlation between the two PGIs (see Eq. (17)). Alternatively, one could also divide the observed standardized polygenic indices $PGI_{1,st}$ and $PGI_{2,st}$ by the same scaling factor:

$$PGI_{1,+} = \frac{PGI_{1,st}}{\frac{\sigma_{PGI^*}}{\sqrt{\sigma_{PGI^*}^2 + \sigma_\nu^2}}} = \frac{\frac{PGI^* + \nu_1}{\sqrt{\sigma_{PGI^*}^2 + \sigma_\nu^2}}}{\frac{\sigma_{PGI^*}}{\sqrt{\sigma_{PGI^*}^2 + \sigma_\nu^2}}} = \frac{PGI^* + \nu_1}{\sigma_{PGI^*}}; \tag{22}$$

$$PGI_{2,+} = \frac{PGI_{2,st}}{\frac{\sigma_{PGI^*}}{\sqrt{\sigma_{PGI^*}^2 + \sigma_\nu^2}}} = \frac{\frac{PGI^* + \nu_2}{\sqrt{\sigma_{PGI^*}^2 + \sigma_\nu^2}}}{\frac{\sigma_{PGI^*}}{\sqrt{\sigma_{PGI^*}^2 + \sigma_\nu^2}}} = \frac{PGI^* + \nu_2}{\sigma_{PGI^*}}. \tag{23}$$

If we then base the IV estimator upon these scaled polygenic indices $PGI_{1,+}$ and $PGI_{2,+}$, then the resulting estimator is given by

$$\hat{\beta}_+^{IV} = \frac{\frac{Cov(Y, PGI_{2,+})}{V(PGI_{2,+})}}{\frac{Cov(PGI_{1,+}, PGI_{2,+})}{V(PGI_{2,+})}} = \frac{Cov\left(\alpha + \beta PGI^* + \varepsilon, \frac{PGI^* + \nu_2}{\sigma_{PGI^*}}\right)}{Cov\left(\frac{PGI^* + \nu_1}{\sigma_{PGI^*}}, \frac{PGI^* + \nu_2}{\sigma_{PGI^*}}\right)} = \beta\frac{\frac{\sigma_{PGI^*}^2}{\sigma_{PGI^*}}}{\frac{\sigma_{PGI^*}^2}{\sigma_{PGI^*}^2}} \tag{24}$$

$$= \beta\sigma_{PGI^*} \equiv \beta_{st}.$$

In sum, an IV estimate of the true standardized effect size can be obtained by (i) dividing the two independent standardized PGIs by the square root of their correlation, and (ii) using these scaled PGIs as instrumental variables for each other. This newly derived scaling factor avoids having to rescale regression estimates ex post to retrieve the estimated heritability as is done for example in ref. 29. In Supplementary Results 3, we show that our ORIV estimates are not biased when applying the between-family correlation between PGIs in the within-family analyses.

Since environments differ, the best linear genetic predictor (i.e., the true latent PGI) may however differ across samples. This would imply that the genetic correlation between the two samples would be lower than 1 for a particular outcome variable. This can be tested, for example with LDSC[73]. LDSC can also be used to estimate the cross-trait LDSC intercept. With a trend towards ever-larger GWAS meta-analyses and reuse of genetic data, it is important to verify that GWAS summary statistics used to create the two independent PGIs are not computed from partially overlapping samples (or from sets of individuals that are related to each other). Using two PGIs based on such related GWAS summary statistics would be in violation of the ORIV assumptions. The cross-trait LDSC intercept can be used to test for sample overlap, although the precise threshold depends on the GWAS samples sizes and the heritability of the trait[74]. In our study, the LDSC cross-trait intercepts do not cross the critical threshold. Moreover, in the UKB split-sample analyses, the GWAS samples do not contain related individuals (see Supplementary Methods 2).

**Bias in IV.** Instrumental Variable regression provides consistent estimates, yet it is biased in small samples with the bias in IV regression inversely related to the first-stage $F$-statistic with a factor roughly equal to $1/F$[42,43]. The $F$-statistic in a univariate regression is equal to the square of the $t$-statistic of the first-stage coefficient $\hat{\tau}$. Therefore, in

this context:

$$\begin{aligned} F = t^2 &= \left[\frac{\hat{\tau}}{s.e.(\hat{\tau})}\right]^2 \\ &= \frac{\hat{\tau}^2}{\left[\frac{\sqrt{1-\hat{\tau}^2}}{\sqrt{N-2}}\right]^2} \\ &= \frac{\hat{\tau}^2(N-2)}{1-\hat{\tau}^2} \\ &= \frac{Corr(PGI_1, PGI_2)^2(N-2)}{1 - Corr(PGI_1, PGI_2)^2}. \end{aligned} \tag{25}$$

Where $\hat{\tau}$ is the first-stage coefficient, and we have used the fact that the PGIs are standardized such that the coefficient $\hat{\tau}$ represents the correlation between the two PGIs. Hence, the performance of (OR)IV depends on the correlation between the two PGIs. This correlation is a function of the measurement error of the independent PGIs (see Eq. (17)). Equation (25) also implies that, like OLS, the performance of (OR)IV is (largely) independent of the absolute values of $\beta$ and $\sigma_{PGI^*}^2$. Unlike OLS, the bias of (OR)IV decreases with the prediction sample size $N$.

## Obviously-related instrumental variables
The most efficient implementation of the proposed IV estimator is the recently proposed technique "Obviously-Related Instrumental Variables" (ORIV) by ref. 31. The idea is to use a "stacked" model

$$\begin{pmatrix} Y \\ Y \end{pmatrix} = \begin{pmatrix} \alpha_1 \\ \alpha_2 \end{pmatrix} + \beta \begin{pmatrix} PGI_{1,+} \\ PGI_{2,+} \end{pmatrix} + \varepsilon, \tag{26}$$

where one instruments the stack of estimated PGIs $\begin{pmatrix} PGI_{1,+} \\ PGI_{2,+} \end{pmatrix}$ with the matrix

$$\begin{pmatrix} PGI_{2,+} & O_N \\ O_N & PGI_{1,+} \end{pmatrix} \tag{27}$$

in which $N$ is the number of individuals and $O_N$ an $N \times 1$ vector with zero's. The implementation of ORIV is straightforward: simply create a stacked dataset and run a Two-Stage Least Squares (2SLS) regression while clustering the standard errors at the individual level. In other words, replicate the dataset creating two values for each individual, and then generate two variables (i.e., an independent variable and an instrumental variable) that alternatively take the value of $PGI_{1,+}$ and $PGI_{2,+}$. The resulting estimate is the average of the estimates that one would get by instrumenting $PGI_{1,+}$ by $PGI_{2,+}$, and vice versa. This procedure makes most efficient use of the information in the two independent PGIs and avoids having to arbitrarily select one PGI as IV for the other. Family fixed effects can also be included in the model, in which case one should include a family-stack fixed effect in order to conduct only within-family comparisons within a stack of the data. Standard errors should then be clustered at both the family as well as the individual level. Supplementary Results 3 illustrates the benefits of applying ORIV over regular IV in terms of point estimates, and a slight improvement in precision. Example syntax in STATA and R is available on our Github webpage https://github.com/geighei/ORIV.

## Within-family analysis
So far, we have ignored the potential influence of parental genotype on the individual's outcome. Controlling for parental genotype is important since the genotype of the child is only truly random conditional on parental genotype. In other words, the only relevant omitted variables in a regression of an outcome on the child's genotype are the genotype of the father and the mother. Leaving parental genotype out is not

innocuous. As evidenced by several studies showing the difference between between-family and within-family analyses[18], the role of parental genotype can be profound. Another way of showing this is by studying the effect of non-transmitted alleles of parents on their children's outcomes[53], to estimate so-called genetic nurture. The true data generating process (DGP) may therefore be:

$$Y = \alpha + \sum_{j=1}^{J} \beta_j^{GWAS} \mathrm{SNP}_j + \sum_{j=1}^{J} \beta_j^{F,GWAS} \mathrm{SNP}_j^F + \sum_{j=1}^{J} \beta_j^{M,GWAS} \mathrm{SNP}_j^M + \varepsilon, \quad (28)$$

where the superscripts $F$ and $M$ denote father and mother, respectively. When the true DGP is governed by Eq. (28), $\beta_j^{GWAS}$ will be estimated with bias in case Eq. (3) is used in a GWAS. A simple solution would be to control for parental genotype or family fixed effects in the GWAS phase. However, with the recent exception of[30], GWAS discovery samples with sufficient parent-child trios or siblings are typically not available. Hence, a researcher often has no option but to work with the "standard GWAS" coefficients that are obtained with Eq. (3) and that produce a biased PGI. Given this empirical reality, it is also the approach we adopt in our simulations.

In a between-family design, the bias in the coefficient of the resulting PGI tends to be upward, as the coefficients of the individuals and his/her parents are typically of the same sign[55]. However, interestingly, when the conventional PGI is used in a within-family design, the bias is downward. The intuition is that when a conventional GWAS (i.e., a GWAS that does not control for parental genotype) is used in its construction, a PGI reflects direct genetic effects as well as indirect genetic effects (e.g., genetic nurture) arising from parental genotype. When applying these PGIs within-families, some of the differences in the PGI across siblings therefore spuriously reflect the effects of parental genotype, whereas in fact their parental genotype is identical. Hence, genetic nurture can be seen as measurement error in the PGI when applied in within-family analyses, leading to an attenuation bias[55].

A final source of downward bias could stem from indirect genetic effects, sometimes referred to as social genetic effects[56,75], e.g., arising from siblings. For example, consider a case with two siblings where there is a direct effect $\gamma_j$ of one's sibling's SNP on the outcome of the other:

$$Y_{1j} = \alpha_j + \sum_{j=1}^{J} \beta_j \mathrm{SNP}_{1j} + \sum_{j=1}^{J} \gamma_j \mathrm{SNP}_{2j} + \varepsilon_{1j}$$

$$Y_{2j} = \alpha_j + \sum_{j=1}^{J} \beta_j \mathrm{SNP}_{2j} + \sum_{j=1}^{J} \gamma_j \mathrm{SNP}_{1j} + \varepsilon_{2j}.$$

When taking sibling differences to eliminate the family fixed effects, we obtain:

$$Y_{1j} - Y_{2j} = \sum_{j=1}^{J} \left( \beta_j - \gamma_j \right) \left( \mathrm{SNP}_{1j} - \mathrm{SNP}_{2j} \right) + \left( \varepsilon_{1j} - \varepsilon_{2j} \right).$$

Since sibling effects are again likely to have the same sign as the direct effect, sibling effects cause a downward bias in the estimated effect of one's own SNP, as measured by $\beta_j$. Again, the only way to overcome this source of bias is to include the parental genotype in the GWAS, since conditional on the parental genotype, the genotypes of siblings are independent.

In sum, within-family analyses are the gold standard to estimate direct genetic effects, free from bias arising from the omission of parental genotype. However, when using a family fixed effects strategy on the basis of a PGI from a conventional GWAS that did not include parental genotype, the direct genetic effect is biased downward as a result of measurement error, genetic nurture effects and social genetic effects[76]. Therefore, this approach provides a lower bound estimate on the direct genetic effects.

## Reporting summary

Further information on research design is available in the Nature Portfolio Reporting Summary linked to this article.

## Data availability

This research has been conducted using the UK Biobank resource (Application number 41382). UK Biobank data can be applied for through https://www.ukbiobank.ac.uk/enable-your-research/apply-for-access. The polygenic indices were constructed using the UK Biobank, the GIANT summary statistics available at https://portals.broadinstitute.org/collaboration/giant/index.php/GIANT_consortium_data_files, and the 23andMe summary statistics. The 23andMe summary statistics are only available to qualified researchers under an agreement with 23andMe that protects the privacy of the 23andMe research participants. For more information, visit https://research.23andme.com/collaborate/#dataset-access/. All results supporting the findings described in this manuscript are available in the article and its Supplementary Information files and from the corresponding author upon request.

## Code availability

All syntax and details on the simulation analyses are available on https://github.com/devlaming/gnames. All syntax for the empirical analyses can be found on https://github.com/geighei/ORIV.

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

## Acknowledgements

The authors gratefully acknowledge participants of the 23andMe, Inc. cohort for sharing GWAS summary statistics for educational attainment. This work made use of the Dutch national e-infrastructure with the support of the SURF Cooperative using grant no. EINF-2327. The authors also acknowledge funding from NORFACE through the Dynamic of Inequality across the Life Course (DIAL) program (462-16-100; H.v.K., P.B., S.v.H., C.A.R., S.F.W.M., D.M., R.D.P.), from the European Research Council (DONNI 851725 to S.v.H and GEPSI 946647 to C.A.R.), from the National Institute on Aging of the National Institutes of Health (RF1055654, R56AG058726 to T.J.G., H.v.K. and RO1AG078522 to T.J.G.), and from the Dutch Research Council (016.VIDI.185.044 to T.J.G.). H.v.K., P.B., T.J.G., S.v.H., and C.A.R. also acknowledge the European Union's Horizon 2021 research and innovation program under the Marie Skłodowska-Curie grant agreement (ESSGN 101073237). This research was supported by the National Institute for Health Research (NIHR Cambridge BRC-1215-20014 for E.A.W.S.). The views expressed are those of the authors and not necessarily those of the National Institutes of Health, NIHR, or the Department of Health and Social Care. We would like to thank Sjoerd van Alten, Dan Belsky, Neil Davies, Ben Domingue, Michel Nivard, and Elliot Tucker-Drob for valuable comments.

## Author contributions

H.v.K and C.A.R. conceived and designed the analysis. R.D.P., S.F.W.M., and D.M. contributed data or analysis tools; H.v.K., E.A.W.S., R.d.V., and C.A.R. performed the simulations and analyzed the results; H.v.K. and C.A.R. performed the empirical analysis; H.v.K., P.B., R.D.P., T.J.G., S.v.H., S.F.W.M., D.M., E.A.W.S., R.d.V., and C.A.R. contributed to writing the paper.

## Competing interests

The authors declare no competing interests.
