## [Peer Review File · Nature Communications]

Overcoming Attenuation Bias in Regressions using Polygenic IndicesReviewers' comments:

Reviewer #1 (Remarks to the Author):

The manuscript describes a method that leverages techniques from the field of instrumental variables and applies them to the analysis of polygenic scores (PGSs). The main application is to adjust for the effect of measurement error in PGSs, which is explored in simulations and in empirical application to height and educational attainment (EA). It's an interesting idea that has some advantages over other approaches to this issue in certain contexts and applications. However, I found the manuscript incomplete in its exploration of the advantages/disadvantages of the proposed approach compared to other approaches, particular the approach of Becker et al. 2021, which is likely to be the main other option researchers will consider.

My primary concern is that the simulations performed are not very realistic, and that there is no direct comparison to the approach of Becker et al. in the simulations. The authors use height and EA as example traits. However, both of these traits are affected by strong assortative mating, a phenomenon that is not considered in this manuscript. The Becker et al. method relies upon GREML to estimate SNP heritability. However, as shown recently, GREML is upwardly biased when assortative mating is present (see <https://doi.org/10.1101/2021.03.18.436091>). I wonder if an advantage of the authors' proposed method is that it would not be biased by assortative mating. It would be useful to see a side-by-side comparison of the authors' proposed method and the method of Becker et. al from realistic simulations under different scenarios: direct effects only, direct effects and genetic nurture, and direct effects and assortative mating. Also, since putting a lower bound on the direct effect heritability is a proposed application of the authors' method and some interesting empirical results on this are presented, this should also be tested in simulations.

Related to this concern, the authors should apply their method to a trait that is not affected by assortative mating, such as heel bone mineral density.

Also, while the current simulations consider the case of unequal measurement error from the two independent PGSs, they do not consider the case of different genetic correlations between the discovery samples and between the discovery and prediction samples.

I also take issue with the way that the authors' method is sometimes described as 'boosting predictive power'. The method does not do this, as far as I can tell: it is designed to enable estimation of parameters in terms of true underlying genetic components, but it does not construct polygenic predictors with higher out-of-sample R^2 than polygenic predictors constructed from meta-analysis estimates. In particular, I think the abstract should specify that the proposed method outperforms meta-

analysis in terms of estimating the coefficient and variance of the true, underlying genetic component, not in terms of out-of-sample R^2 between genetic predictor and phenotype.

Minor comments:

I found the introduction to be excessively long and somewhat repetitive with the results and methods sections. Also, readers from human genetics may not be familiar with instrumental variable methods, so a couple of sentences introducing IV methods to the reader could be helpful.

'For EA, it takes a sample of ~ 1 million to construct a PGS explaining 0.20' -- the authors are using a theoretical prediction here that is contradicted by the empirical results from GWAS of EA, so perhaps better not to put this explicitly in terms of EA

'Our paper also closely relates to' -> 'is closely related to'

'will also disattenuate PGS estimates in case no or very imprecise external information' -- this is clumsy, consider rephrasing

I also found the second sentence of the third paragraph of page 4 starting 'First, when the prediction sample' to be difficult to parse.

'genetic nurture' should be defined when it is first introduced

First sentence after equation 4, 'GWAS-based heritability on basis' delete 'on basis'

Page 7, I am not so sure the Becker et al. approach has zero bias: see above issue with assortative mating.

Figure 2 and similar figures: I found it hard to distinguish the different lines, consider using colors.

Line 6 of first paragraph of 'Empirical Illustrations': 'on basis' -> 'on the basis'

Figure 4: I found it confusing that the x-axes do not start at 0

Methods: I found use of the 'hat' (caret) over the beta for the expected standardized PGS coefficient to be confusing. Usually, this indicates a sample estimator, whereas here it is being used to denote a population parameter.

Start of 'obviously related instrumental variables' subsection of methods: this might be better suited to the introduction.

Start of paragraph with equation (29): 'Unlike OLS, the performance of (OR)IV improves with the prediction sample size'. Without defining performance here, this seems to imply OLS estimators don't get better with increasing sample size, but sampling error will decrease, so they do get better by that metric.

'a case with two siblings when there is a direct effect of one's sibling's SNP' -- shouldn't this be an indirect genetic effect?

Supplement: GWAS: how were individuals of non-European ancestry filtered?

Also, fast-GWA was applied to samples that have been pruned of relatives. Since fast-GWA constructs a sparse-GRM with non-zero elements for individuals with relatedness above 0.05, it is probable that the estimates from fast-GWA here are almost identical to OLS. While I don't think this is an issue, the authors would be better off applying BOLT-LMM if they want to take advantage of the boost in power that comes from modeling genome-wide SNP effects as random-effects.

Reviewer #2 (Remarks to the Author):

In this study, the authors propose an instrument variable approach to increase predictive power of polygenic scores. They present theory, perform simulations and present results from empirical analyses on educational attainment (EA) and height. The authors' main claim is that their IV-based method improves prediction accuracy compared to current ('conventional') practice. Overall, I found the paper confusing, because the authors frame the problem (and solution) in terms of regression coefficients rather than in correlations between improved genetic predictors and outcome. They claim to improve

out-of-sample prediction accuracy (e.g., in the Title) but do not seem to provide empirical evidence that prediction accuracy is increased when using IV.

1. Bias definition and random effects models. To maximise the accuracy of out-of-sample prediction, the estimates of SNP effects from GWAS should be based upon the true (but unknown) distribution of SNP effects (e.g., Goddard et al. 2009, *Statistical Science*). This implies applying random effects models, where OLS SNP effect sizes are shrunk towards zero. If the assumptions about the prior distribution of effect sizes are met, then the predicted PGS (\hat{g}) are unbiased in the sense that $E(y|\hat{g}) = E(g|\hat{g}) = \hat{g}$, and the correlation between y and \hat{y} ($= \hat{g}$) is maximised. In other words, in a linear regression of phenotype on genetic predictor, the expected slope is 1.0 (the variance of \hat{g} is equal to the covariance between g and \hat{g}). This kind of random effects predictor has an easy and intuitive interpretation: the difference in outcome (future observation) of any 2 individuals is, on average, equal to their difference in genetic predictor. The authors instead use a traditional fixed effects (OLS) definition of unbiasedness, which is $E(\hat{g}) = g$. This leads to all kinds of scaling and re-scaling issues. Becker et al. 2021 go through the same process as in the current paper, because they also use a fixed-effects framework, which needs post-hoc scaling to account of 'measurement' error. I note that LDpred is actually based upon a random effects model, and therefore produces shrunk estimates of SNP effects. Indeed, if the infinitesimal model assumption (an option in LDpred) is correct then the resulting estimates of \hat{g} have best linear unbiased predictor (BLUP) properties. BLUP was first proposed by an econometrician (Goldberger 1962, *JASA*; see also Robinson 1991, *Statistical Science*). In the current paper, the authors standardise the PGS from LDpred, which seems odd since it is, in my view, already on the correct scale.

2. Sufficient statistics. If OLS estimates of SNPs effects in a GWAS are sufficient statistics to derive polygenic scores from, then how can splitting up the data in two halves increase the amount of information (and thereby the prediction accuracy)?

3. State-of-the-art in prediction. The authors propose that the conventional way of boosting predictive power is to increase sample size. I agree that increasing sample size is very important, not least because asymptotically it will lead to estimated SNP effects (and thereby estimated PGS) without error. However, there is a large amount of literature, ignored by the authors, where researchers have used Bayesian methods with external information to inform on prior distributions of SNP effect sizes to improve prediction accuracy (e.g., Zhang et al. 2021, *Nature Communications*, and references therein). These methods clearly improve prediction accuracy when compared to either OLS estimates of SNP effects or in comparison to infinitesimal priors. My suggestion is that the authors acknowledge this development, even if the social sciences have been slow to capitalise on them.

4. Theory. If we write the model in Eq(6) as $y = g + e$, and $\text{var}(y) = 1$, then from Eq(8), $\beta = 1$. This is very interpretable in my view, i.e. one unit in g changes y with one unit. Eq(10): This is confusing because this is a regression of y on the true additive factor, so $R^2 = h^2$ (with $h^2 = \text{SNP-heritability}$).

Usually, R^2 (or its expectation, see page 2) is from a regression of y on the estimated (predicted) value of g , not the true value. The expected R^2 given on page 2 is from OLS estimation of SNP effects, and is not the correct equation if a random effects model is used (e.g., when using LDpred) – see e.g. Pasaniuc & Price 2017 (Nature Reviews Genetics).

5. Empirical results. I hope that I am not misunderstanding the results in Table 1 & 2, but the authors present their results as regression coefficients (or their squares) and not the achieved correlation between predictor and outcome in out-of-sample prediction? For the OLS columns, the ‘ h^2_{GWAS} ’ results are equal to the regression R^2 if y and x are standardised to unit variance. However, the IV approach gives, as I understand it, a “predicted R^2 ” and not an actual correlation between polygenic predictor and the trait? The results between OLS(UK) and ORIV(Split) seem very large and I wonder if they are just the result of scaling.

Let M = number of SNPs, N = sample size and h^2 = SNP-heritability. If the SNPs are independent and their effect estimated with OLS then $\text{var}(\hat{g}) = \text{var}(g) + \text{var}(v) = h^2 + M/N$, and for out-of-sample prediction, $\text{beta}(y \text{ on } \hat{g}) = h^2/[h^2 + M/N]$. If $h^2 = 0.5$, $M = 100,000$ and $N = 300,000$ (similar to height in UKBB) then $\text{beta} = 0.6$ and the prediction accuracy $R^2 = 0.3$. If \hat{g} is standardised to unit variance, the $\text{beta}(y \text{ on } \hat{g}^*) = h^2/\sqrt{h^2+M/N} = 0.548$.

Now we split N into two with sample sizes $N_1 + N_2 = N$, estimate SNP effects using OLS from both datasets, creating \hat{g}_1 and \hat{g}_2 . $\text{var}(\hat{g}_i) = h^2 + M/N_i$, with $i=1,2$ and $\text{cov}(\hat{g}_1, \hat{g}_2) = h^2$ (Eq. 17). Note that in general the correlation of \hat{g}_1 and \hat{g}_2 depends on their variances, which are not necessarily the same, and the denominator in Eq(18) seems wrong. In my notation, the denominator is $\sqrt{(h^2 + M/N_1)(h^2 + M/N_2)}$. This is only equal to $\text{var}(g) + \text{var}(v)$ if $N_1=N_2=N/2$ and if $\text{var}(v) = M/(N/2) = 2M/N$.

Rescaling, using Eq(25) of the authors, gives $E(\text{beta}_{\text{IV}}) = h = \sqrt{h^2} = 0.71$. Although this shows that $0.71 > 0.548$, it doesn’t mean that we have improved prediction power, we have just rescaled the estimates. As far as I can understand it, the numbers in this hypothetical example are equivalent to the reported 0.579 and 0.625 in the first row of Table 1.

6. I don’t think that we need a new parameter (“ h^2_{GWAS} ”) or a new term (“explained SNP-based heritability”), in particular if its expectation is the regression R^2 from an actual regression analysis of y on \hat{g} . Eq(4) can be written as R^2/h^2 .

7. If the genetic correlation between the true latent values in two discovery samples/populations (g_1 and g_2 in my notation) are less than 1 then the ‘best’ genetic predictor depends on which target population is used. Eq(17) can be generalised to $rg^* \sqrt{h^2_{1} * h^2_{2}}$, with h^2_i the SNP-heritability in

population i and r_g the genetic correlation. If, for example, we are interested in out-of-sample prediction in UKBB and we have discovery data from UKBB and 23andMe, and if $r_g < 1$, then the PGS derived from the UKBB sample should get more weight than that derived from 23andMe. It is my understanding the authors use some kind of average PGS in their IV-derived regression coefficient. Multi-trait prediction methods like MTAG can handle such heterogeneity ($r_g < 1$). Even if $r_g = 1$ but h^2_1 is not equal to h^2_2 then instrumenting both PGS and averaging them would not seem optimal. My suggestion is that the authors discuss under what circumstances instrumenting both PGS and taking an average would lead to bias.

8. Please can the authors provide more information on the PGS they use from LDpred, e.g. what the model is and how many SNPs were used?

Overcoming Attenuation Bias in Regressions using Polygenic Indices: A Comparison of Approaches

September 12, 2022

Response to reviewer 1

The manuscript describes a method that leverages techniques from the field of instrumental variables and applies them to the analysis of polygenic scores (PGSs). The main application is to adjust for the effect of measurement error in PGSs, which is explored in simulations and in empirical application to height and educational attainment (EA). It's an interesting idea that has some advantages over other approaches to this issue in certain contexts and applications. However, I found the manuscript incomplete in its exploration of the advantages/disadvantages of the proposed approach compared to other approaches, particular the approach of Becker et al. 2021, which is likely to be the main other option researchers will consider.

Response: We appreciate your words and thank you for sharing your time and expertise. Your incisive comments have helped us further improve and sharpen the focus of our paper. In particular, we reframed our study to the comparison of three approaches to estimate the effect of a polygenic index (i.e., meta-analyzing, ORIV, and the PGI repository correction (PGI-RC) suggested by Becker et al. 2021).

1. *My primary concern is that the simulations performed are not very realistic, and that there is no direct comparison to the approach of Becker et al. in the simulations. The authors use height and EA as example traits. However, both of these traits are affected by strong assortative mating, a phenomenon that is not considered in this manuscript. The Becker et al. method relies upon GREML to estimate SNP heritability. However, as shown recently, GREML is upwardly biased when assortative mating is present (see <https://doi.org/10.1101/2021.03.18.436091>). I wonder if an advantage of the authors' proposed method is that it would not be biased by assortative mating. It would be useful to see a side-by-side comparison of the authors' proposed method and the method of Becker et. al from realistic simulations under different scenarios:*

direct effects only, direct effects and genetic nurture, and direct effects and assortative mating. Also, since putting a lower bound on the direct effect heritability is a proposed application of the authors' method and some interesting empirical results on this are presented, this should also be tested in simulations.

Response: Point well taken. Following your suggestion, we developed a completely new simulation tool called GNAMEs (Genetic Nurture and Assortative Mating Effect Simulator) to conduct a comprehensive set of simulations with scenarios that vary in:

- The GWAS discovery sample size;
- The prediction sample size;
- Genetic nurture;
- Assortative mating;
- Genetic correlation.

Moreover, by simulating 2 children per family, the simulation set-up also allows to compare between- and within-family estimates in all these scenarios. We believe the new set of results is considerably more realistic and enables a fair comparison across the different approaches to prevent attenuation bias in the regression coefficients of a PGI.

The results show that the PGI-RC performs well and better than ORIV when the genetic correlation between the discovery and prediction sample is low, or when the GWAS discovery sample is small. However, as you anticipated, the PGI-RC is particularly vulnerable to assortative mating and here ORIV clearly does better. Moreover, in within-family analyses, where the PGI-RC cannot be applied, we confirm that ORIV is able to tighten the lower bound on the estimation of direct genetic effects. Please find a more detailed description of the results in the completely re-written section on Simulation results.

2. *Related to this concern, the authors should apply their method to a trait that is not affected by assortative mating, such as heel bone mineral density.*

Response: Your suggestion to also analyze a trait (alongside height and educational attainment) that is not affected by assortative mating is excellent. Assortative mating is widespread across human traits, and bone mineral density is indeed well-known as a trait for which assortative mating is virtually absent. As our UKB data application did not cover bone mineral density, we instead analyzed diastolic blood pressure. While assortative mating on this trait is non-zero, it is very low (Robinson et al., 2017), providing a useful test case to verify that assortative mating is not driving our main conclusions. Our results confirm the findings for EA and height, and are presented in Supplementary Information E.1.

3. *Also, while the current simulations consider the case of unequal measurement error from the two independent PGSs, they do not consider the case of different genetic correlations between the discovery samples and between the discovery and prediction samples.*

Response: Agreed. In the new simulation set-up we allow for imperfect genetic correlation between the two GWAS discovery samples, as well as imperfect genetic correlation between (one of the) discovery samples and the prediction sample. The results can be found in Figure 5 in the paper: the PGI-RC is largely insensitive to imperfect genetic correlation, while ORIV is especially sensitive to an imperfect genetic correlation between the discovery and prediction sample, and less sensitive to an imperfect genetic correlation between the two discovery samples used to construct the independent PGIs.

4. *I also take issue with the way that the authors' method is sometimes described as 'boosting predictive power'. The method does not do this, as far as I can tell: it is designed to enable estimation of parameters in terms of true underlying genetic components, but it does not construct polygenic predictors with higher out-of-sample R^2 than polygenic predictors constructed from meta-analysis estimates. In particular, I think the abstract should specify that the proposed method outperforms meta-analysis in terms of estimating the coefficient and variance of the true, underlying genetic component, not in terms of out-of-sample R^2 between genetic predictor and phenotype.*

Response: Thank you for raising this important point. We fully agree, and the revised manuscript does not mention 'boosting predictive power' anymore: we instead focus on the correct estimation of the effect of the polygenic index, with ORIV and the PGI-RC as approaches to prevent attenuation bias in regression coefficients rather than boosting the predictive power of a PGI. We have also changed the title of the paper to avoid confusion. In the introduction we now explicitly mention: "Our goal is to estimate a coefficient that is free from attenuation bias due to measurement error in a PGI. As such, we are not primarily interested in boosting the out-of-sample predictive power of a given PGI (in terms of e.g., the R-squared) for which ever-increasing GWAS discovery samples remain important."

Minor comments

5. *I found the introduction to be excessively long and somewhat repetitive with the results and methods sections. Also, readers from human genetics may not be familiar with instrumental variable methods, so a couple of sentences introducing IV methods to the reader could be helpful.*

Response: The introduction of the revised manuscript has been rewritten drastically. As a result, it has become considerably shorter, while still being able to provide some intuition on IV methods.

6. *'For EA, it takes a sample of 1 million to construct a PGS explaining 0.20' – the authors are*

using a theoretical prediction here that is contradicted by the empirical results from GWAS of EA, so perhaps better not to put this explicitly in terms of EA.

Response: Agreed. We softened the emphasis on EA on making this point.

7. *'Our paper also closely relates to' -> 'is closely related to'*

Response: Changed.

8. *'will also disattenuate PGS estimates in case no or very imprecise external information' – this is clumsy, consider rephrasing*

Response: Done.

9. *I also found the second sentence of the third paragraph of page 4 starting 'First, when the prediction sample' to be difficult to parse.*

Response: We have rephrased this sentence.

10. *'genetic nurture' should be defined when it is first introduced*

Response: Done.

11. *First sentence after equation 4, 'GWAS-based heritability on basis' delete 'on basis'*

Response: Done.

12. *Page 7, I am not so sure the Becker et al. approach has zero bias: see above issue with assortative mating.*

Response: Agreed: The new simulations clearly show that in case of assortative mating and genetic nurture, the PGI-RC of Becker et al. does not provide correct estimates of the effect of the additive genetic factor.

13. *Figure 2 and similar figures: I found it hard to distinguish the different lines, consider using colors.*

Response: Point taken. We have replaced all previous figures to simpler coefficient plots, while including the corresponding Root Mean Squared Errors in tables in the Supplementary Information (SI).

14. *Line 6 of first paragraph of 'Empirical Illustrations': 'on basis' -> 'on the basis'*

Response: Done.

15. *Figure 4: I found it confusing that the x-axes do not start at 0*

Response: We have updated all figures in the revised manuscript. For visual clarity however, we have sometimes omitted the 0 if the estimated coefficients are all in the range of e.g., 0.5.

16. *Methods: I found use of the 'hat' (caret) over the beta for the expected standardized PGS coefficient to be confusing. Usually, this indicates a sample estimator, whereas here it is being used to denote a population parameter.*

Response: We see the potential confusion as you are right that our derivations are not specific to any sample. Still, in our notation we would like to distinguish between an estimated coefficient and the true coefficient. Given that we use regression tools, the “hat” notation helps to make clear the distinction between an estimated regression coefficient and the true population value. Because of your comment, we experimented with alternative ways of notation such as using asterisks. However, these alternatives all have their own disadvantages (e.g., they often interfere with other types of notations such as exponents). Therefore, we made it clearer in the revised manuscript and SI that the “hat” notation refers to an estimated coefficient and kept the notation as it was. If you have other specific suggestions how to make that distinction clearer, we’re happy to adopt it.

17. *Start of 'obviously related instrumental variables' subsection of methods: this might be better suited to the introduction.*

Response: Thank you for this suggestion. While we left the details for the Appendix, we now discuss ORIV more extensively in the introduction.

18. *Start of paragraph with equation (29): 'Unlike OLS, the performance of (OR)IV improves with the prediction sample size'. Without defining performance here, this seems to imply OLS estimators don't get better with increasing sample size, but sampling error will decrease, so they do get better by that metric.*

Response: Agreed and we have rewritten as follows: “Unlike OLS, the bias of (OR)IV decreases with the prediction sample size N.”

19. *'a case with two siblings when there is a direct effect of one's sibling's SNP' – shouldn't this be an indirect genetic effect?*

Response: Yes, sibling effects are also indirect genetic effects. We clarified terminology in the revised manuscript.

20. *Supplement: GWAS: how were individuals of non-European ancestry filtered?*

Response: To focus on individuals of European ancestry only, we exclude individuals with a value larger than 0 on the first principal component of the genetic data. We also filtered based on the self-reported ethnicity. Specifically, we only include individuals if their ethnic background is "White", "British", "Irish", or "Any other white background". We included these selection criteria in the revised SI.

21. *Also, fast-GWA was applied to samples that have been pruned of relatives. Since fast-GWA constructs a sparse-GRM with non-zero elements for individuals with relatedness above 0.05, it is probable that the estimates from fast-GWA here are almost identical to OLS. While I don't think this is an issue, the authors would be better off applying BOLT-LMM if they want to take advantage of the boost in power that comes from modeling genome-wide SNP effects as random-effects.*

Response: In the GWAS discovery sample, we removed genetically related individuals from the prediction sample (i.e., the sibling subsample). However, there are still some genetically related individuals within the GWAS discovery sample left. For this reason, we use the fastGWA routine to control for genetic relatedness in the GWAS analysis. We clarified this in the revised SI.

In closing, we want to thank you again for providing all the excellent comments that have helped us to implement meaningful changes to our study. We hope you are satisfied with the revised manuscript!

Response to reviewer 2

In this study, the authors propose an instrument variable approach to increase predictive power of polygenic scores. They present theory, perform simulations and present results from empirical analyses on educational attainment (EA) and height. The authors' main claim is that their IV-based method improves prediction accuracy compared to current ('conventional') practice. Overall, I found the paper confusing, because the authors frame the problem (and solution) in terms of regression coefficients rather than in correlations between improved genetic predictors and outcome. They claim to improve out-of-sample prediction accuracy (e.g., in the Title) but do not seem to provide empirical evidence that prediction accuracy is increased when using IV.

Response: Thank you for reading and commenting extensively upon our manuscript. Your detailed and constructive recommendations have been very useful for improving our study. We can see where your confusion arose, and the revised manuscript does not mention 'boosting predictive power' anymore. We instead focus on the correct estimation of the effect of the polygenic index, and ORIV and the PGI repository correction (PGI-RC) as approaches to overcome attenuation bias in regression coefficients rather than boosting the predictive power of a PGI.

1. *Bias definition and random effects models. To maximise the accuracy of out-of-sample prediction, the estimates of SNP effects from GWAS should be based upon the true (but unknown) distribution of SNP effects (e.g., Goddard et al. 2009, Statistical Science). This implies applying random effects models, where OLS SNP effect sizes are shrunk towards zero. If the*

assumptions about the prior distribution of effect sizes are met, then the predicted PGS (\hat{g}) are unbiased in the sense that $E(y|\hat{g}) = E(g|\hat{g}) = \hat{g}$, and the correlation between y and \hat{g} ($= g$) is maximised. In other words, in a linear regression of phenotype on genetic predictor, the expected slope is 1.0 (the variance of \hat{g} is equal to the covariance between g and \hat{g}). This kind of random effects predictor has an easy and intuitive interpretation: the difference in outcome (future observation) of any 2 individuals is, on average, equal to their difference in genetic predictor. The authors instead use a traditional fixed effects (OLS) definition of unbiasedness, which is $E(\hat{g}) = g$. This leads to all kinds of scaling and re-scaling issues. Becker et al. 2021 go through the same process as in the current paper, because they also use a fixed-effects framework, which needs post-hoc scaling to account of ‘measurement’ error. I note that LDpred is actually based upon a random effects model, and therefore produces shrunk estimates of SNP effects. Indeed, if the infinitesimal model assumption (an option in LDpred) is correct then the resulting estimates of \hat{g} have best linear unbiased predictor (BLUP) properties. BLUP was first proposed by an econometrician (Goldberger 1962, JASA; see also Robinson 1991, Statistical Science). In the current paper, the authors standardise the PGS from LDpred, which seems odd since it is, in my view, already on the correct scale.

Response: Thank you for sharing these insightful thoughts. PGIs are increasingly available to applied researchers in a number of rich longitudinal datasets of particular relevance to epidemiologists and social scientists. Therefore, these applied researchers will typically not construct PGIs themselves but they do face the question how to obtain a correct estimate of the effect of the PGI. As suggested by Reviewer 1, we now frame the paper as a comparison between the PGI-RC suggested by Becker et al. (2021) and ORIV. Since (i) both estimators are fixed effects estimators and target an unbiased, or at least consistent, estimator of the PGI coefficient in a regression context, and (ii) the applications we cite in the introduction on e.g., GxE interactions or testing mechanisms through which the PGI influences an outcome, also typically adopt an OLS regression framework, we feel that it’s natural and valuable to adopt the same practice here. We included this motivation in the revised introduction.

2. *Sufficient statistics. If OLS estimates of SNPs effects in a GWAS are sufficient statistics to derive polygenic scores from, then how can splitting up the data in two halves increase the amount of information (and thereby the prediction accuracy)?*

Response: We can see where the confusion came from and realize that our previous focus on ‘prediction accuracy’ was misplaced and confusing. In fact, you are right that the prediction accuracy (in terms of e.g., R-squared) of the split-sample PGIs is lower. The intuition of why splitting up the data helps is as follows. If one splits the GWAS discovery sample, one obtains two PGIs that both proxy the same underlying “true” PGI. In the case of no measurement error, theoretically the correlation between the two PGIs should approach 1. However, in

practice there exists measurement error and the empirical correlation will be below 1. The IV approach then infers the amount of measurement error through the empirical correlation between the two split-sample PGIs, and in turn uses this information to correct (or ‘scale’) the observed association between the PGI and the phenotype.

In more formal terms, as derived in equation (6) in the manuscript, in the case of measurement error, the attenuation in the OLS coefficient is given by:

$$\hat{\beta} = \beta \frac{\sigma_{PGI*}^2}{\sigma_{PGI*}^2 + \sigma_v^2},$$

where β is the coefficient of the true PGI, σ_{PGI*}^2 is the variance of the true PGI, and σ_v^2 is the variance of measurement error. Clearly, the attenuation factor is unobserved. However, as we derive in equation (16) in the manuscript, under certain assumptions, the correlation between two independent PGIs is exactly equal to this attenuation factor:

$$Corr(PGI_1, PGI_2) = \frac{\sigma_{PGI*}^2}{\sigma_{PGI*}^2 + \sigma_v^2}.$$

Hence, by scaling a reduced form coefficient of a PGI on the outcome by the correlation between two independent PGIs gives a consistent effect of the coefficient of the true PGI. This is exactly what the IV approach does using ‘two-stage least squares’. Hence, even though the prediction accuracy of the two split-sample PGIs is lower, and even though IV does not deliver a PGI that is more predictive of the outcome out-of-sample, it is able to retrieve a consistent estimate of the coefficient of the true PGI if the IV assumptions are met.

We have conveyed this intuition more extensively in the paper.

3. *State-of-the-art in prediction. The authors propose that the conventional way of boosting predictive power is to increase sample size. I agree that increasing sample size is very important, not least because asymptotically it will lead to estimated SNP effects (and thereby estimated PGS) without error. However, there is a large amount of literature, ignored by the authors, where researchers have used Bayesian methods with external information to inform on prior distributions of SNP effect sizes to improve prediction accuracy (e.g., Zhang et al. 2021, Nature Communications, and references therein). These methods clearly improve prediction accuracy when compared to either OLS estimates of SNP effects or in comparison to infinitesimal priors. My suggestion is that the authors acknowledge this development, even if the social sciences have been slow to capitalise on them.*

Response: Agreed, this is a fair point. In our view, the goal of these studies is slightly different. Zhang et al. (2020) take a given GWAS and explore which priors and settings of constructing a PGI lead to the highest predictive accuracy. Marquez-Luna et al. (2021)

incorporate functional priors and show how this can enhance the predictive power of a PGI. Both approaches hold the level of classical measurement error constant and make the most out of the GWAS discovery sample by using the best construction method. Our goal instead is to overcome the disattenuation induced by measurement error, while holding the construction method of a PGI constant.

Nonetheless, we take your point that our representation of the state-of-the-art was not entirely correct. We now acknowledge in the revised introduction that there are important advances in how PGIs are constructed to enhance the predictive power of PGIs.

4. *Theory. If we write the model in Eq(6) as $y = g + e$, and $\text{var}(y) = 1$, then from Eq(8), $\beta = 1$. This is very interpretable in my view, i.e. one unit in g changes y with one unit. Eq(10): This is confusing because this is a regression of y on the true additive factor, so $R^2 = h^2$ (with $h^2 = \text{SNP-heritability}$). Usually, R^2 (or its expectation, see page 2) is from a regression of y on the estimated (predicted) value of g , not the true value. The expected R^2 given on page 2 is from OLS estimation of SNP effects, and is not the correct equation if a random effects model is used (e.g., when using LDpred) – see e.g. Pasaniuc & Price 2017 (Nature Reviews Genetics).*

Response: Point taken. To avoid confusion, we have now dropped both the equation on the expected R-squared from page 2, as well as the derivation in the methods on the R-squared of the true additive factor (formerly equation 10).

5. *Empirical results. I hope that I am not misunderstanding the results in Table 1 & 2, but the authors present their results as regression coefficients (or their squares) and not the achieved correlation between predictor and outcome in out-of-sample prediction? For the OLS columns, the 'h2GWAS' results are equal to the regression R^2 if y and x are standardised to unit variance. However, the IV approach gives, as I understand it, a "predicted R^2 " and not an actual correlation between polygenic predictor and the trait? The results between OLS(UK) and ORIV(Split) seem very large and I wonder if they are just the result of scaling.*

Let $M = \text{number of SNPs}$, $N = \text{sample size}$ and $h^2 = \text{SNP-heritability}$. If the SNPs are independent and their effect estimated with OLS then $\text{var}(g\text{-hat}) = \text{var}(g) + \text{var}(v) = h^2 + M/N$, and for out-of-sample prediction, $\beta(y \text{ on } g\text{-hat}) = h^2 / [h^2 + M/N]$. If $h^2 = 0.5$, $M = 100,000$ and $N = 300,000$ (similar to height in UKBB) then $\beta = 0.6$ and the prediction accuracy $R^2 = 0.3$. If $g\text{-hat}$ is standardised to unit variance, the $\beta(y \text{ on } g\text{-hat}^) = h^2 / \sqrt{h^2 + M/N} = 0.548$.*

Now we split N into two with sample sizes $N1 + N2 = N$, estimate SNP effects using OLS from both datasets, creating $g1\text{-hat}$ and $g2\text{-hat}$. $\text{var}(gi\text{-hat}) = h^2 + M/Ni$, with $i=1,2$ and $\text{cov}(g1\text{-hat}, g2\text{-hat}) = h^2$ (Eq. 17). Note that in general the correlation of $g1\text{-hat}$ and $g2\text{-hat}$

depends on their variances, which are not necessarily the same, and the denominator in Eq(18) seems wrong. In my notation, the denominator is $\sqrt{(h^2 + M/N1)(h^2 + M/N2)}$. This is only equal to $\text{var}(g) + \text{var}(v)$ if $N1=N2=N/2$ and if $\text{var}(v) = M/(N/2) = 2M/N$.

Rescaling, using Eq(25) of the authors, gives $E(\text{betaIV}) = h = \sqrt{h^2} = 0.71$. Although this shows that $0.71 > 0.548$, it doesn't mean that we have improved prediction power, we have just rescaled the estimates. As far as I can understand it, the numbers in this hypothetical example are equivalent to the reported 0.579 and 0.625 in the first row of Table 1.

Response: Your interpretation is correct. We focus on reducing attenuation bias in the regression coefficients, not the predictive power of a PGI out-of-sample. Hence, IV does not provide an actual R-squared, but we report an implied heritability on basis of the estimated squared standardized coefficient. You are also correct in asserting that in deriving equation (16, formerly 18) we assumed that the variance of measurement error over the variance of the true PGI are the same in the two PGIs (i.e., $N1=N2=N/2$), which holds automatically when splitting the GWAS sample into two random halves. Inspired by your comments, we have now made it clear in the title and introduction that our goal is to disattenuate regression coefficients, not to increase the predictive power of a PGI.

6. *I don't think that we need a new parameter ("h2GWAS") or a new term ("explained SNP-based heritability"), in particular if its expectation is the regression R^2 from an actual regression analysis of y on $g\text{-hat}$. Eq(4) can be written as R^2/h^2 .*

Response: Point taken, we no longer use these terms in the revised manuscript.

7. *If the genetic correlation between the true latent values in two discovery samples/populations ($g1$ and $g2$ in my notation) are less than 1 then the 'best' genetic predictor depends on which target population is used. Eq(17) can be generalised to $rg \times \sqrt{h2_1 \times h2_2}$, with $h2_i$ the SNP-heritability in population i and rg the genetic correlation. If, for example, we are interested in out-of-sample prediction in UKBB and we have discovery data from UKBB and 23andMe, and if $rg < 1$, then the PGS derived from the UKBB sample should get more weight than that derived from 23andMe. It is my understanding the authors use some kind of average PGS in their IV-derived regression coefficient. Multi-trait prediction methods like MTAG can handle such heterogeneity ($rg < 1$). Even if $rg = 1$ but $h2_1$ is not equal to $h2_2$ then instrumenting both PGS and averaging them would not seem optimal. My suggestion is that the authors discuss under what circumstances instrumenting both PGS and taking an average would lead to bias.*

Response: This is a very interesting suggestion. A weighted IV estimator would probably most easily be estimated in a genomic Structural Equation Model (SEM) where one can flexibly allow the factor loadings to be different across the two measurements of a PGI. A full comparison of the PGI repository correction versus ORIV versus a weighted ORIV or genomic SEM is

beyond the scope of this paper, but we have mentioned this as a promising direction for future research in the Discussion section.

Inspired by your comment, we also investigated how imperfect genetic correlation influences ORIV using extensive simulations. The results are presented in Figure 5. In brief, when there exists imperfect genetic correlation between the discovery and prediction sample, both meta-analysis and ORIV are biased, with the degree of bias inversely related to the genetic correlation. In these cases, we now advocate using the PGI-RC suggested by Becker et al. (2021) since this approach uses the SNP-heritability estimated in the prediction sample for rescaling. Interestingly, when there exists imperfect genetic correlation between the two discovery samples, yet not between one of the discovery samples and the prediction sample, then the performance of ORIV is remarkably good. Even when the genetic correlation between the two discovery samples is ‘only’ 0.75 (far lower than what we observe between, e.g., 23andMe and UKB for Educational Attainment) then ORIV produces consistent estimates.

8. *Please can the authors provide more information on the PGS they use from LDpred, e.g. what the model is and how many SNPs were used?*

In the revised SI, we included additional details about the constructed PGIs (including a table with the GWAS discovery sample sizes and the number of SNPs included in each PGI).

In closing, we appreciate the time you have invested in providing us with excellent comments. We hope that our revisions are in line with your expectations!

References

Becker, J., Burik, C. A., Goldman, G., Wang, N., Jayashankar, H., Bennett, M., ... & Okbay, A. (2021). Resource profile and user guide of the Polygenic Index Repository. *Nature Human Behaviour*, 5(12), 1744-1758.

Liu, L., Zhao, M., Xie, Z. G., Liu, J., Peng, H. P., Pei, Y. F., ... & Zhang, L. (2020). Twelve new genomic loci associated with bone mineral density. *Frontiers in Endocrinology*, 11(1), 243.

Márquez-Luna, C., Gazal, S., Loh, P. R., Kim, S. S., Furlotte, N., Auton, A., & Price, A. L. (2021). Incorporating functional priors improves polygenic prediction accuracy in UK Biobank and 23andMe data sets. *Nature Communications*, 12(1), 1-11.

Robinson, M. R., Kleinman, A., Graff, M., Vinkhuyzen, A. A., Couper, D., Miller, M. B., ... & Visscher, P. M. (2017). Genetic evidence of assortative mating in humans. *Nature Human Behaviour*,

1(1), 1-13.

Zhang, Q., Sidorenko, J., Couvy-Duchesne, B., Marioni, R. E., Wright, M. J., Goate, A. M., ... & Visscher, P. M. (2020). Risk prediction of late-onset Alzheimer's disease implies an oligogenic architecture. *Nature Communications*, 11(1), 1-11.

REVIEWER COMMENTS

Reviewer #1 (Remarks to the Author):

The revised manuscript is greatly improved in terms of both presentation and results. I have a few outstanding comments, but the manuscript should help researchers who wish to account for measurement error in their PGI analyses.

Abstract: lower bounds on the direct effect are given, but without units.

Simulation study: it says SNPs are simulated to be independent, but they won't be independent under assortative mating

Formula for h^2_{SNP} with genetic nurture: this is only accurate when direct and indirect effects are uncorrelated. While this is stated in the methods/supplement, I think it should be stated also in the main text

I am not convinced that ORIV is biased under assortative mating. I think ORIV is actually doing better than the authors believe. I think what is changing is the true coefficient, or the true amount of variance explained by true PGI (measured without noise). To simplify discussion, consider a phenotype affected by direct effects only, then the variance explained by the true PGI in a random mating population is v_g , but for assortative mating at equilibrium, the variance explained is $v_g/(1-r)$, where r is the equilibrium correlation between parents PGIs. This means that the correct coefficient for a PGI standardized to have variance 1 is actually larger than $\sqrt{v_g}$. Similarly, I think in your simulation with genetic nurture, the correct coefficient is not 0.5 when there is assortative mating: it should be the square root of the R^2 between phenotype and PGI, which will be greater the stronger assortative mating is.

Discussion: 'estimated direct effect to be around 3.5%' again please give the units/interpretation of this 'effect'

Equation 12 in methods: I don't think the Delta method is necessary here

Equation 14: shouldn't there be a hat on both the h^2_{SNP} and R^2 terms too since these are sample estimates in real world applications?

Reviewer #3 (Remarks to the Author):

This manuscript investigates the performance of two methods to reduce bias in regression coefficient of phenotypes on polygenic indices (PGI). Through extensive simulations and real data analysis, the authors find both methods, ORIV and PGI-RC, to be better than the common practice of using meta-analysed PGI in terms of bias, which is defined as the deviate to the slope of regressing the phenotypes on the standardised PGI. Between ORIV and PGI-RS, they find that one could be better than the other depending on the GWAS sample size, prediction sample size, and the presence of indirect genetic effect or assortative mating. By comparing ORIV results from within-family and between-family analyses, they detect substantial indirect genetic effects for educational attainment but not much for height.

I believe that their theory and analysis are correct under a fixed effect model. However, I agree with Reviewer 2 that there should be no bias under a random effect model, provided that the assumption on the prior distribution for SNP effects is correct. If I understand correctly, the problem of bias as described in the paper in fact results from a scaling issue due to the standardisation of PGI from LDpred which is a random effect model. I would prefer to describe this as a rescaling issue rather than biasness (see below). The authors should at least discuss and clarify this in the paper to avoid confusion.

All PGI in the analysis are constructed using LDpred with a prior value of 1. Presumably, they mean the mixing probability equals to 1 so it is a BLUP model where all SNPs contribute to phenotypic variation with a normal prior distribution. This SNP effect model is equivalent to the individual effect model, $y = g + e$, where g is PGI considered as random, and the BLUP estimate for g is $g_{BLUP} = c'V^{-1}y$, where c is the covariance between y and g (genomic relationship between training individuals and the individual to be predicted), V is the variance of y (Henderson 1975 Biometrics). According to the property of BLUP, the regression slope should be 1, i.e.,

$$\text{Slope} = \text{cov}(y, g_{BLUP})/\text{var}(g_{BLUP}) = 1$$

$$\text{because } \text{cov}(y, g_{BLUP}) = \text{cov}(g+e, c'V^{-1}y) = c'V^{-1}c \text{ and } \text{var}(g_{BLUP}) = \text{var}(c'V^{-1}y) = c'V^{-1}V V^{-1}c = c'V^{-1}c.$$

This indicates that g_{BLUP} (PGI from LDpred) is on the same scale of y (regardless of y is standardised or not) and is unbiased. Now if we rescale g_{BLUP} by its standard deviation, i.e., $g_{BLUP_rescaled} = g_{BLUP}/\sqrt{c'V^{-1}c} = c'V^{-1}y/\sqrt{c'V^{-1}c}$, then the slope will become

$$\text{Slope} = \text{cov}(y, g_{BLUP_rescaled}) = c'V^{-1}c/\sqrt{c'V^{-1}c} = \sqrt{c'V^{-1}c}$$

which is not equal to 1 but can be predicted from theory because both c and V are known. Thus, I believe the “bias” described in their paper is actually a rescaling issue and the scale factor can be easily calculated according to the BLUP theory.

GWAS samples are often affected by unobserved confounding factors, such as population stratification. In this case the ORIV method would be biased because the residual correlation between PGI of the two separate samples would not be zero. The authors should discuss the potential impact of uncorrected confounding factors in GWAS samples on the two methods presented in the paper.

p5 bottom, "One notable exception is that for a small prediction sample size ($N \leq 1,000$), the PGI-RC slightly underestimates the true coefficient, because the estimated GREML SNP-heritability that is used in the PGI-RC is measured with noise in such a small sample." If the GREML estimated SNP-heritability is unbiased, should the coefficient estimate not be biased as well? In addition, it is more often to apply GREML to the GWAS data set which should have a larger sample size, therefore this bias is unlikely to be substantial in practice but it is important to understand the bias.

Minor:

p2, "For example, for a trait with a SNP-heritability of 25%, it takes a sample of ~1 million to construct a PGI explaining 20%, but it would take a 7-fold increase in discovery sample size to achieve an R2 of 24%." Please give a reference how it is calculated.

p3, it seems the last two paragraphs should belong to Results and Discussion sections, respectively.

Eq (14): should the denominator be $2 \cdot h^2_{\text{SNP}}$?

Why Table 2 and 3 do not show prediction R2 result for ORIV and PGI-RC?

Response Letter

Overcoming Attenuation Bias in Regressions using Polygenic Indices: A Comparison of Approaches

April 6, 2023

Response to Reviewer 1

The revised manuscript is greatly improved in terms of both presentation and results. I have a few outstanding comments, but the manuscript should help researchers who wish to account for measurement error in their PGI analyses.

Response: Many thanks for your constructive and helpful comments throughout the process. Much appreciated.

1. *Abstract: lower bounds on the direct effect are given, but without units.*

Response: These are standardized (i.e., both the outcome and the PGI are standardized to have mean 0 and standard deviation 1) effect sizes. We have added the word ‘standardized’ to the abstract for clarification.

2. *Simulation study: it says SNPs are simulated to be independent, but they won't be independent under assortative mating*

Response: Point well taken. We now better clarify in the paper that we simulate the SNP *effect sizes* to be independent. But, you are correct that under assortative mating there will be a covariance structure across the SNPs as well as across the direct and indirect genetic effects. We now discuss this in detail in the simulation scenario with assortative mating (page 8 of the revised manuscript as well as in the newly added Supplementary Information A8 and A9).

3. *Formula for h_{SNP}^2 with genetic nurture: this is only accurate when direct and indirect effects are uncorrelated. While this is stated in the methods/supplement, I think it should be stated*

also in the main text

Response: Thank you for this comment. We have added “If the direct genetic effect and genetic nurture components are independent, the SNP-based heritability is given by: $h_{SNP}^2 = h^2 + 0.5n^2$ ” to the main text (see page 4). This now explicitly clarifies to the reader that the formula relies on this assumption.

4. *I am not convinced that ORIV is biased under assortative mating. I think ORIV is actually doing better than the authors believe. I think what is changing is the true coefficient, or the true amount of variance explained by true PGI (measured without noise). To simplify discussion, consider a phenotype affected by direct effects only, then the variance explained by the true PGI in a random mating population is v_g , but for assortative mating at equilibrium, the variance explained is $v_g/(1 - r)$, where r is the equilibrium correlation between parents PGIs. This means that the correct coefficient for a PGI standardized to have variance 1 is actually larger than $\sqrt{v_g}$. Similarly, I think in your simulation with genetic nurture, the correct coefficient is not 0.5 when there is assortative mating: it should be the square root of the R^2 between phenotype and PGI, which will be greater the stronger assortative mating is.*

Response: We thank the reviewer for this astute observation. The short answer is that you were completely right, and we are grateful that you pushed us to dig deeper into this. The longer answer with further analyses follows below.

For our simulation design with both genetic nurture (GN) and assortative mating (AM) we observe three key points in Figure 4 of the main text:

- (a) for between-family analyses, both ORIV and REML-based approaches produce larger coefficients than $\sqrt{h^2 + \frac{1}{2}n^2} = \sqrt{0.2 + \frac{1}{2} \times 0.1} = \sqrt{0.25} = 0.5$, where h^2 (resp. n^2) denotes the proportion of phenotypic variance accounted for by direct genetic effects (genetic nurture effects) of SNPs under the assumption of independence between the direct genetic effects and genetic nurture;
- (b) the higher the level of AM, the higher the coefficients,
- (c) REML-based approaches produce higher coefficients than ORIV;

After careful consideration, we believe there are two convoluted issues shaping these results: (i) standardisation of genetic liabilities in each generation; and (ii) the interplay between GN and AM. We now consider each of these issues in isolation, to clarify its impact on the results.

Issue 1: Standardisation. Given that we work with a finite number of SNPs (i.e., $M \rightarrow \infty$), the infinitesimal model (i.e., infinitely many SNPs, each with an infinitesimally small effect,

yet potentially aggregating to infinite variances, e.g., when $r = 1$ in your formula) is only an approximation: the true polygenic index (PGI) without standardisation will always have a finite variance, even under perfect AM once equilibrium has been reached.

Bearing this in mind, we can safely standardise the true PGI (in terms of the direct contribution of SNPs) to unit variance and mean zero. This type of transformation is exactly what happens in each generation of our forward simulation. For instance, in the absence of GN, the standardised PGI is assigned coefficient $\sqrt{h^2}$ and the error term (also with mean zero and unit variance) is assigned coefficient $\sqrt{1 - h^2}$, and thus the PGI explains proportion h^2 of the phenotypic variance for the new generation, irrespective of the degree of AM, and irrespective of whether equilibrium has been reached or not.

Therefore, in the absence of GN, we expect ORIV to yield a consistent estimate $\sqrt{h^2}$ in our simulation, regardless of (the degree of) AM. To support this claim, we carried out the same type of simulation as for which results are reported in Figure 4, but now setting $h^2 = 0.25$ and $n^2 = 0$ (i.e., no GN). Indeed, Figure 1 in this response letter (which has also been added to Figure 4 in the main text) shows that for all levels of AM, the ORIV estimates do not differ significantly from the true coefficient, which equals $\sqrt{0.25} = 0.5$, both when applied between family (left panel) and within family (right panel). In contrast, REML-based estimates differ significantly from the expected coefficient in line with earlier results [2].

Issue 2: The interplay between GN and AM. We now turn our attention to the real culprit: emergent correlation between the direct and indirect components under our design combining GN and AM, which is possible even when the SNP effects shaping the direct and indirect components are independent (see Supplementary Information A8 for an extensive intuition for why this correlation emerges). Under our design combining AM and GN, the proportion of phenotypic variance accounted for by the direct and indirect components combined equals

$$\frac{h^2 + n^2 + 2\rho_{g,n}\sqrt{h^2n^2}}{1 + 2\rho_{g,n}\sqrt{h^2n^2}}, \quad (1)$$

and the expected coefficient equals

$$\sqrt{\frac{h^2 + \frac{1}{2}n^2 + 2\rho_{g,n}\sqrt{h^2n^2}}{1 + 2\rho_{g,n}\sqrt{h^2n^2}}}, \quad (2)$$

where $\rho_{g,n}$ denotes the correlation between the direct and indirect components (see Supplementary Information A9 for derivations). Therefore, as you correctly point out, this changes the definition of our target parameter.

Figure 1: Estimated coefficients for the Polygenic Index (PGI) and their corresponding 95% confidence intervals based on between-family analyses (left-panel) and within-family analyses (right-panel) using meta-analysis (circles), Obviously Related Instrumental Variables (ORIV, rhombuses), the default PGI-RC procedure (squares), and the PGI-RC procedure taking into account uncertainty in the GREML estimates (triangles) in a scenario without genetic nurture and varying levels of assortative mating (AM). The simulations hold constant the Genome-wide Association Study (GWAS) discovery sample size such that the resulting meta-analysis PGI has an R^2 of 15.4%, and the prediction sample size is held fixed at $N = 16,000$. The dashed line represents the true coefficient. The simulation results are based on 100 replications.

Since a positive value for $\rho_{g,n}$ in our simulations arises naturally in a scenario with AM and GN (we did not fix $\rho_{g,n}$ ourselves, it is produced in the simulations), we cannot compare our coefficient estimates directly to Equation 2 by plugging in the simulated values for h^2 , n^2 , and $\rho_{g,n}$. Instead, to gauge the performance of ORIV and the PGI-RC in this scenario, we empirically estimate the implied correlation $\hat{\rho}_{g,n}$ from our 100 simulation runs and plug this estimate into Equation 2.

Table 1 in this response letter shows a summary of the simulation results. When both genetic nurture and assortative mating are present, the target parameter changes due to the arising positive correlation between direct and indirect genetic effects. In this case, ORIV continues to approximate the target parameter very well, whereas the PGI-RC is biased. We have included these results and their discussion in the revised manuscript (page 8 in the main text, Figure 4 in the main text, and Table 9 in Supplementary Information B).

5. *Discussion: ‘estimated direct effect to be around 3.5%’ again please give the units/interpretation of this ‘effect’*

Response: We have added the word ‘standardized’ to the description and have converted the percentages into effect sizes: “On the basis of within-family analyses, in our application,

Table 1: Results accompanying Figure 4c in the main text.

AM	$\hat{\rho}_{g,n}$	Expected coefficient	ORIV	PGI-RC
0	-0.002	0.500	0.497	0.500
0.25	0.037	0.508	0.506	0.518
0.50	0.080	0.516	0.515	0.536
0.75	0.132	0.526	0.529	0.555
1.00	0.185	0.536	0.538	0.576

Notes: Expected and estimated coefficients for the Polygenic Index (PGI) using Obviously Related Instrumental Variables (ORIV) and the PGI-RC procedure in a scenario with varying levels of assortative mating (AM) and genetic nurture. $\hat{\rho}_{g,n}$ denotes the empirically observed correlation between the direct and indirect genetic component. The simulations hold constant the Genome-wide Association Study (GWAS) discovery sample such that the resulting meta-analysis PGI has an R^2 of 15.4%, and the prediction sample size is held fixed at $N = 16,000$. The simulation results are based on 100 replications.

ORIV estimated the standardized direct genetic effect to be around 0.18 for EA and 0.6 for height, respectively a 30% (EA) and 14% (height) increase compared with a meta-analysis PGI.”

6. *Equation 12 in methods: I don't think the Delta method is necessary here*

Response: Agreed. We have deleted the reference to the Delta method here.

7. *Equation 14: shouldn't there be a hat on both the h_{SNP}^2 and R^2 terms too since these are sample estimates in real world applications?*

Response: Agreed. We have added a hat to these terms in Equation 14 (Equation 15 in the revised manuscript).

In closing, we want to thank you again for providing all these excellent comments. They have helped us implement meaningful changes to our study. We hope you are satisfied with the revised manuscript!

Response to Reviewer 3

We would like to thank Reviewer 3 for carefully studying our paper, and for important and instructive comments that have helped to improve the paper.

1. *I believe that their theory and analysis are correct under a fixed effect model. However, I agree with Reviewer 2 that there should be no bias under a random effect model, provided that*

the assumption on the prior distribution for SNP effects is correct. If I understand correctly, the problem of bias as described in the paper in fact results from a scaling issue due to the standardisation of PGI from LDpred which is a random effect model. I would prefer to describe this as a rescaling issue rather than biasness (see below). The authors should at least discuss and clarify this in the paper to avoid confusion.

All PGI in the analysis are constructed using LDpred with a prior value of 1. Presumably, they mean the mixing probability equals to 1 so it is a BLUP model where all SNPs contribute to phenotypic variation with a normal prior distribution. This SNP effect model is equivalent to the individual effect model, $y = g + e$ where g is PGI considered as random, and the BLUP estimate for g is $g_{BLUP} = c'V^{-1}y$, where c is the covariance between y and g (genomic relationship between training individuals and the individual to be predicted), V is the variance of y (Henderson 1975 Biometrics). According to the property of BLUP, the regression slope should be 1, i.e.,

$$\text{Slope} = \text{cov}(y, g_{BLUP}) / \text{var}(g_{BLUP}) = 1$$

because $\text{cov}(y, g_{BLUP}) = \text{cov}(g + e, c'V^{-1}y) = c'V^{-1}c$ and $\text{var}(g_{BLUP}) = \text{var}(c'V^{-1}y) = c'V^{-1}VV^{-1}c = c'V^{-1}c$. This indicates that g_{BLUP} (PGI from LDpred) is on the same scale of y (regardless of y is standardised or not) and is unbiased. Now if we rescale g_{BLUP} by its standard deviation, i.e., $g_{BLUP, rescaled} = g_{BLUP} / \text{sqrt}(c'V^{-1}c) = c'V^{-1}y / \text{sqrt}(c'V^{-1}c)$, then the slope will become

$$\text{Slope} = \text{cov}(y, g_{BLUP, rescaled}) = c'V^{-1}c / \text{sqrt}(c'V^{-1}c) = \text{sqrt}(c'V^{-1}c)$$

which is not equal to 1 but can be predicted from theory because both c and V are known. Thus, I believe the “bias” described in their paper is actually a rescaling issue and the scale factor can be easily calculated according to the BLUP theory.

Response: Under a Gaussian infinitesimal prior, the LDpred PGI and the BLUP are, indeed, closely related. As stated in the original LDpred paper: “Interestingly, under these assumptions the resulting effects approximate the standard mixed-model genomic BLUP effects. LDpred-inf is therefore a natural extension of the genomic BLUP to summary statistics” [9]. The key word, however, is *approximate*. Since a PGI (whether constructed using LDpred or any other technique) is constructed on the basis of coefficients from a finite-sample GWAS, there is measurement error in these coefficients and hence the PGI is subject to the same type of measurement error. The resulting attenuation bias of the PGI in a regression of an outcome on the PGI is therefore not a scaling issue induced by standardization of the PGI, but rather

an inherent ‘bias’ stemming from measurement error in the GWAS coefficients.

Another reason to use the word ‘bias’ is that our focus is on two methods that reduce attenuation bias in regression analyses involving PGIs: PGI-RC and ORIV. In the application of these methods, it is typical to use the corresponding fixed effects (OLS) definition of bias because both estimators are fixed effects estimators and target an unbiased, or at least consistent, estimator of the PGI coefficient in a regression context. The applications we cite in the introduction on, e.g., G×E interactions or on the testing of mechanisms through which the PGI influences an outcome, also typically adopt an OLS regression framework. For example, when one is interested in how the EA PGI influences wealth accumulation [e.g., 1], one cannot rely on BLUP or GREML, but one has to rely on an empirically constructed PGI. For these reasons, we feel that it is natural and valuable to adopt a fixed effects setting in our study and use the terminology of ‘attenuation bias’, which is the common term used in the econometrics literature. We have clarified in the Introduction what we mean by attenuation bias: “Empirically constructed PGIs are nonetheless a noisy proxy for the true (latent) PGI because, amongst other reasons, the GWAS underlying the construction of the PGI is based on a finite sample [3, 5]. The noise in the GWAS coefficients translates into noisy measures of the PGI and leads to what is typically known as ‘attenuation bias’ (i.e., a bias towards zero) in regressions.” Further, in the Methods section we clarified that we adopt a fixed effect model.

Having said that, we do agree that attenuation bias can also be seen as a rescaling problem in a regression setting, where due to measurement error we obtain an attenuated coefficient estimate. In fact, the methods we study here, PGI-RC and ORIV, both rescale the coefficient to obtain the true coefficient. In the paper we acknowledge this on pages 2 and 3 in the context of IV (“The IV approach uses this information to correct (or ‘scale’) the observed association between the PGI and the outcome”), and in the context of the PGI-RC (“Intuitively, in this approach the SNP-based heritability is estimated in a first step, after which the coefficient of the PGI is re-scaled to match this SNP-based heritability”).

2. *GWAS samples are often affected by unobserved confounding factors, such as population stratification. In this case the ORIV method would be biased because the residual correlation between PGI of the two separate samples would not be zero. The authors should discuss the potential impact of uncorrected confounding factors in GWAS samples on the two methods presented in the paper.*

Response: Consider the data generating process $Y = \alpha + \beta PGI^* + \varepsilon$, where the error term ε captures the effects of confounding variables not controlled for in the model. In case not all confounding variables are controlled for, the OLS estimator of β should not be understood as the causal effect of PGI^* , but as capturing all the effects of PGI^* and its correlates. Therefore, in a typical GWAS with a limited set of control variables the coefficient of the PGI is likely

to reflect both direct and indirect genetic effects. Similarly, the h_{SNP}^2 estimate of GREML also captures all effects that can be traced back to genetic variation across individuals in the sample, which is likely to include both direct and indirect genetic effects.

In the Introduction we now write: “We are also not primarily interested in estimating the direct (or ‘causal’) effect of a PGI. PGIs typically capture not only effects of inherited variation (direct effects), but also effects of demography and relatives [so-called indirect genetic effects 6]. Therefore, when correcting for measurement error in a between-family design, the target parameter is the coefficient of the additive SNP-factor, which encompasses both direct as well as indirect genetic effects. In a within-family design, the target parameter is the direct genetic effect.”

Your question may however additionally relate to whether insufficient controlling for confounding variables in the GWAS model would not only affect the interpretation of ORIV (and PGI-RC) estimates but also bias them. In ORIV, we use two independent GWAS samples to obtain two noisier proxies for PGI^* : $PGI_1 = PGI^* + \nu_1$ and $PGI_2 = PGI^* + \nu_2$. To have $Cov(\nu_1, \nu_2) = 0$ at the PGI level, *estimation errors* in β_j^{GWAS} need to be independent at the SNP level across the two GWAS samples. It has been derived theoretically and shown using simulations by the developers of GIV regression [4] that the condition $Cov(\nu_1, \nu_2) = 0$ generally holds *for polygenic traits* (i.e., traits influenced by many genetic variants, each with a very small effect). The reason for this is that polygenic scores are constructed using a very large number of SNPs, and correlations in estimation errors at the SNP-level cancel each other out upon aggregation to the PGI level. In the revised manuscript, we now mention this assumption explicitly in the Introduction: “In case (i) of polygenicity, (ii) the sources of measurement error are independent, and (iii) the relative variance of measurement error in the PGI is the same across the two discovery samples, then the correlation between the two PGIs reveals the degree of measurement error. The IV approach in turn uses this information to correct (or ‘scale’) the observed association between the PGI and the outcome.”

Finally, whereas we do not model population stratification in our simulations explicitly, we do model genetic nurture and assortative mating at the SNP level. These population phenomena have very similar effects as those of population stratification on the genotype-phenotype relationship [7]. We now write on page 8 of the paper: “Since the results with and without genetic nurture are very similar, we confirm the earlier finding that confounding factors at the GWAS stage do not bias the ORIV and PGI-RC estimates in targeting the additive SNP factor between families [4].”

3. *p5 bottom, “One notable exception is that for a small prediction sample size ($N \leq 1,000$), the PGI-RC slightly underestimates the true coefficient, because the estimated GREML SNP-heritability that is used in the PGI-RC is measured with noise in such a small sample.” If the*

GREML estimated SNP-heritability is unbiased, should the coefficient estimate not be biased as well? In addition, it is more often to apply GREML to the GWAS data set which should have a larger sample size, therefore this bias is unlikely to be substantial in practice but it is important to understand the bias.”

Response: We agree with your point about the consistency of the GREML estimate, and therefore removed the sentence “because the estimated GREML SNP-heritability that is used in the PGI-RC is measured with noise in such a small sample” from the text.

We also agree that it’s true that GREML estimates are fairly widely available for many discovery samples. However, if a researcher is using a prediction sample that is rather different in terms of context/environment, then the researcher may be tempted to obtain a SNP-based heritability estimate from the prediction sample rather than relying on an external SNP-based heritability estimate from a completely different context. What we show here is that this approach tends to provide imprecise and, in our case, downward biased estimates of the SNP-based heritability (and hence the PGI-RC estimator) in case the prediction sample is rather small. The revised sentence, on page 5, now reads: “Judging from the point estimates, in a between-family setting, ORIV and the PGI-RC clearly outperform a meta-analysis PGI, with limited differences between them in this scenario. An increasing prediction sample size shrinks the confidence intervals but leaves the coefficients largely unaffected. One notable exception is that for a small prediction sample size ($N \leq 1,000$), the PGI-RC slightly underestimates the true coefficient.”

Minor:

4. *p2, “For example, for a trait with a SNP-heritability of 25%, it takes a sample of ~ 1 million to construct a PGI explaining 20%, but it would take a 7-fold increase in discovery sample size to achieve an R^2 of 24%.” Please give a reference how it is calculated.”*

Response: We have now added a reference to equation (1) in the paper which gives an approximation of the expected R -squared for a PGI for a given SNP-based heritability and sample size.

5. *p3, it seems the last two paragraphs should belong to Results and Discussion sections, respectively.*

Response: In line with your suggestion we have added these paragraphs to the Results and Discussion section, respectively.

6. Eq (14): should the denominator be $2 * h_{SNP}^2$?

Response: We have added the subscript to Equation 14 (Equation 15 in the revised manuscript) in the main text. On basis of the Delta method, we derived for that any x

$$\begin{aligned} V(\sqrt{x}) &= \left(\frac{1}{2}x^{-\frac{1}{2}}\right)^2 V(x) \\ &= \frac{1}{4}x^{-1}V(x) \\ &= \frac{V(x)}{4x} \end{aligned} \tag{3}$$

and so the standard error is given by

$$s.e.(\sqrt{x}) = \frac{s.e.(x)}{2\sqrt{x}} \tag{4}$$

If we plug in $x = h_{SNP}^2$ it follows that

$$s.e.(\sqrt{h_{SNP}^2}) = \frac{s.e.(h_{SNP}^2)}{2h_{SNP}} \tag{5}$$

7. Why Table 2 and 3 do not show prediction R2 result for ORIV and PGI-RC?

Response: In Instrumental Variables (IV) models, the R -squared is an inappropriate measure of fit and not easily compared to OLS measures of fit [e.g., 8]. Moreover, since the PGI-RC method simply scales the PGI coefficient to the square root of the SNP-based heritability, we also felt it is less meaningful to include the R -squared since it is equal to the SNP-based heritability by construction. We now write in the Results section: “The ORIV standardized effect estimate is 0.337. Whereas the R^2 of an IV regression is not meaningful [e.g., 8], we can estimate the implied heritability \hat{h}_{SNP}^2 by squaring the standardized coefficient, which gives 11.4%.”

References

- [1] D. Barth, N. W. Papageorge, and K. Thom. Genetic endowments and wealth inequality. *Journal of Political Economy*, 128(4):1474–1522, 2020.
- [2] R. Border, S. O’Rourke, T. de Candia, M. E. Goddard, P. M. Visscher, L. Yengo, M. Jones, and M. C. Keller. Assortative mating biases marker-based heritability estimators. *Nature Communications*, 13(1):660, 2022.

- [3] H. D. Daetwyler, B. Villanueva, and J. A. Woolliams. Accuracy of predicting the genetic risk of disease using a genome-wide approach. *PLoS ONE*, 3(10):e3395, 2008.
- [4] T. A. DiPrete, C. A. P. Burik, and P. D. Koellinger. Genetic instrumental variable regression: Explaining socioeconomic and health outcomes in nonexperimental data. *Proceedings of the National Academy of Sciences*, 115(22):E4970–E4979, 2018.
- [5] F. Dudbridge. Power and predictive accuracy of polygenic risk scores. *PLoS Genetics*, 9(3):e1003348, 2013.
- [6] L. J. Howe, M. G. Nivard, T. T. Morris, A. F. Hansen, H. Rasheed, Y. Cho, G. Chittoor, R. Ahlskog, P. A. Lind, T. Palviainen, et al. Within-sibship genome-wide association analyses decrease bias in estimates of direct genetic effects. *Nature Genetics*, 54(5):581–592, 2022.
- [7] T. T. Morris, N. M. Davies, G. Hemani, and G. D. Smith. Population phenomena inflate genetic associations of complex social traits. *Science Advances*, 6(16):eaay0328, 2020.
- [8] M. H. Pesaran and R. J. Smith. A generalized R^2 criterion for regression models estimated by the instrumental variables method. *Econometrica*, 62(3):705–710, 1994.
- [9] B. J. Vilhjalmsón, J. Yang, H. K. Finucane, A. Gusev, S. Lindström, S. Ripke, G. Genovese, P.-R. Loh, G. Bhatia, R. Do, T. Hayeck, H.-H. Won, S. Kathiresan, M. Pato, C. Pato, R. Tamimi, E. A. Stahl, N. Zaitlen, B. Pasaniuc, G. Belbin, E. E. Kenny, M. H. Schierup, P. L. De Jager, N. A. Patsopoulos, S. A. McCarroll, M. J. Daly, S. M. Purcell, D. I. Chasman, B. M. Neale, M. Goddard, P. M. Visscher, P. Kraft, N. Patterson, and A. L. Price. Modeling linkage disequilibrium increases accuracy of polygenic risk scores. *American Journal of Human Genetics*, 97(4):576–592, 2015.

REVIEWER COMMENTS

Reviewer #1 (Remarks to the Author):

The authors have done a good job in extending their treatment of assortative mating with direct and indirect genetic effects.

However, I am still a little confused as to why ORIV gives the same estimate regardless of the level of assortative mating. If it was estimating the unstandardized β , as defined in Equation (5), this would make sense. But if it is estimating the standardized coefficient, $\beta_{st} = SD(PGI) * \beta$, as in Equation (6), this would not be true since this would change with the standard deviation of the true PGI, which would increase with assortative mating.

However, on page 3 at the start of the simulation section, the authors state they are estimating the standardized coefficient. And in the response to my previous review, they state "the standardised PGI is assigned coefficient $\sqrt{h^2}$...", and thus the PGI explains proportion h^2 of the phenotypic variance for the new generation, irrespective of the degree of AM, and irrespective of whether equilibrium has been reached or not." By which I think they mean the true standardized PGI has regression coefficient equal to the square root of the heritability in the random-mating population, no matter the degree of assortative mating. This is false since this implies that the variance explained by the true PGI is $(\sqrt{h^2})^2 * 1 = h^2$, constant across generations, no matter the degree of assortment, which contradicts standard theory on assortative mating showing that the genetic variance and heritability is increased. In other words, the true regression coefficient for the true PGI, standardized to have variance 1, should increase with the degree of assortment. (However, the unstandardized coefficient would stay the same, since the AM induced increase in genetic variance is captured by the increase in variance of the true PGI, holding the coefficient constant.)

Can the authors please clarify this matter?

Reviewer #3 (Remarks to the Author):

I appreciate the authors' effort to improve the manuscript and believe that my comments have been adequately addressed.

Response Letter

Overcoming Attenuation Bias in Regressions using Polygenic Indices: A Comparison of Approaches

May 9, 2023

Response to Reviewer 1

The authors have done a good job in extending their treatment of assortative mating with direct and indirect genetic effects.

Response: Many thanks for your constructive and helpful comments throughout the process. Much appreciated.

1. *However, I am still a little confused as to why ORIV gives the same estimate regardless of the level of assortative mating. If it was estimating the unstandardized β , as defined in Equation (5), this would make sense. But if it is estimating the standardized coefficient, $\beta_{st} = SD(PGI) \times \beta$, as in Equation (6), this would not be true since this would change with the standard deviation of the true PGI, which would increase with assortative mating.*

*However, on page 3 at the start of the simulation section, the authors state they are estimating the standardized coefficient. And in the response to my previous review, they state "the standardised PGI is assigned coefficient $\sqrt{h^2}$...", and thus the PGI explains proportion h^2 of the phenotypic variance for the new generation, irrespective of the degree of AM, and irrespective of whether equilibrium has been reached or not." By which I think they mean the true standardized PGI has regression coefficient equal to the square root of the heritability in the random-mating population, no matter the degree of assortative mating. This is false since this implies that the variance explained by the true PGI is $(\text{sqrt}(h^2))^2 * 1 = h^2$, constant across generations, no matter the degree of assortment, which contradicts standard theory on assortative mating showing that the genetic variance and heritability is increased. In other words, the true regression coefficient for the true PGI, standardized to have variance 1, should increase*

with the degree of assortment. (However, the unstandardized coefficient would stay the same, since the AM induced increase in genetic variance is captured by the increase in variance of the true PGI, holding the coefficient constant.)

Can the authors please clarify this matter?

Response: We thank the reviewer for the astute observation. We acknowledge that we have been insufficiently clear on this matter.

In our simulations, the standardisation pertains to the data-generating process (DGP) of the phenotype itself. We designed our simulations such that the true coefficient does not change with increasing levels of assortative mating (AM). To this purpose, we standardise the true PGI to mean zero and unit variance in each generation of the simulation, and then assign effect $\sqrt{h^2}$ to this standardised PGI, when drawing Y as a linear function of this PGI and the other terms in the model. Similarly, the genetic nurture (GN) component is standardised in the same manner in each generation, and then assigned weight $\sqrt{n^2}$.

This approach keeps the ‘contemporary’ heritability fixed (with the exception of the design in which AM and GN are combined, where the ensuing correlation between the true PGI and nurture component confounds our true coefficient). Another way to think about our standardisation is that, under AM and the resulting excess homozygosity rates, inflation in the variance of the PGI is avoided by shrinking all SNP effect sizes by the same factor in a given generation.

Although this approach may be somewhat nonstandard, the reason why we implemented it is that this simplifies our formulation of the true coefficient and, hence, gives a clear and intuitive benchmark in all simulation scenarios (again with the exception of the design in which AM and GN are combined). Table 1 shows the true coefficients for the different simulation scenarios.

Inspired by the reviewer’s comment, we now also analyzed an alternative implementation without standardisation of the variance in the PGI in each generation. In this scenario, as the reviewer rightly points out, the true coefficient evolves over generations, as the panmictic heritability (i.e., h_0^2 in the notation of Border et al. [1]) evolves towards the equilibrium heritability (i.e., h_∞^2). However, to the best of our knowledge, the ‘contemporary’ heritability in a given generation cannot be expressed as a simple function of h_0^2 , h_∞^2 , and the number of generations passed since the population switched from random mating to AM. Hence, there is no tractable expression available for the true coefficient in that case. Instead, in the simulations without per-generation standardisation in the DGP, we regress the outcome on the true PGI in the prediction sample to obtain an accurate and unbiased estimate of the true between-family coefficient. In addition, to estimate the true within-family coefficient in that scenario, we use the same regression, but then also controlling for family-specific fixed effects.

		Between-family		Within-family	
		Standardisation		Standardisation	
GN	AM	Yes	No	Yes	No
No	No	$\sqrt{h^2}$	$\sqrt{h^2}$	$\sqrt{h^2}$	$\sqrt{h^2}$
No	Yes	$\sqrt{h^2}$	$\sqrt{\text{'contemporary' } h^2}$	$\sqrt{h^2}$	$\sqrt{\text{'contemporary' } h^2}$
Yes	No	$\sqrt{h^2 + 0.5n^2}$	$\sqrt{h^2 + 0.5n^2}$	$\sqrt{h^2}$	$\sqrt{h^2}$
Yes	Yes	Eq. 94 in the SI	intractable	$\sqrt{h^2}$	$\sqrt{\text{'contemporary' } h^2}$

Table 1: True coefficient in between-family and within-family analyses for each data-generating process in the simulations, comprising three main settings: (i) genetic nurture (GN), (ii) assortative mating (AM), and (iii) standardisation of the genetic contributions towards the phenotypes in each generation. Note that appropriately weighted offspring genotypes can only tag half of the GN variance, and that the combination of GN and AM leads to a correlated direct and indirect component, considerably complicating the definition of the true coefficient. h^2 denotes the panmictic SNP-based heritability and n^2 the panmictic proportion of variance accounted for by GN effects of SNPs. ‘contemporary’ h^2 is defined as the proportion of phenotypic variance accounted for by the true PGI in a given generation.

Figure 1 presents the results of these new simulations. As the reviewer anticipated, the true standardised coefficient increases with increasing levels of assortative mating in the absence of standardisation. Still, as in our original simulation runs, ORIV provides consistent estimates of the true coefficient, whereas the PGI-RC overestimates the true coefficient when assortative mating is present.

We now discuss the standardisation in Footnote 2 in the revised manuscript, we added the precise data-generating process without such standardisation to our description of GNAMEs in the revised Appendix A.1, and we present the new simulation results without standardisation in the newly added Appendix C.3. We hope these revisions clarify matters adequately.

References

- [1] R. Border, S. O’Rourke, T. de Candia, M. E. Goddard, P. M. Visscher, L. Yengo, M. Jones, and M. C. Keller. Assortative mating biases marker-based heritability estimators. *Nature Communications*, 13(1):660, 2022.

Figure 1: Estimated coefficients for the Polygenic Index (PGI) and their corresponding 95% confidence intervals based on between-family analyses (left-panel) and within-family analyses (right-panel) using meta-analysis (circles), Obviously Related Instrumental Variables (ORIV, rhombuses), the default PGI-RC procedure (squares), and the PGI-RC procedure taking into account uncertainty in the GREML estimates (triangles) in a scenario without standardisation of the genetic contribution towards the phenotypes in each generation, without genetic nurture, and with varying levels of assortative mating. The simulations hold constant the Genome-wide Association Study (GWAS) discovery sample such that the resulting meta-analysis PGI has an R^2 of 15.4%, and the prediction sample size is held fixed at $N = 16,000$. The dashed line represents the true coefficient, which increases with higher levels of assortative mating in the absence of standardisation in each generation. The simulation results are based on 100 replications.

REVIEWERS' COMMENTS

Reviewer #1 (Remarks to the Author):

The authors have cleared up the remaining issue and improved the manuscript further in response.